# CD103$^+$ cDC1 and endogenous CD8$^+$ T cells are necessary for improved CD40L-overexpressing CAR T cell antitumor function

Nicholas F. Kuhn[1,2,3], Andrea V. Lopez[2], Xinghuo Li[2], Winson Cai[2], Anthony F. Daniyan[2] & Renier J. Brentjens [2✉]

While effective in specific settings, adoptive chimeric antigen receptor (CAR) T cell therapy for cancer requires further improvement and optimization. Our previous results show that CD40L-overexpressing CAR T cells mobilize endogenous immune effectors, resulting in improved antitumor immunity. However, the cell populations required for this protective effect remain to be identified. Here we show, by analyzing Batf3$^{-/-}$ mice lacking the CD103$^+$ conventional dendritic cell type 1 (cDC1) subpopulation important for antigen cross-presentation, that CD40L-overexpressing CAR T cells elicit an impaired antitumor response in the absence of cDC1s. We further find that CD40L-overexpressing CAR T cells stimulate tumor-resident CD11b$^-$CD103$^-$ double-negative (DN) cDCs to proliferate and differentiate into cDC1s in wild-type mice. Finally, re-challenge experiments show that endogenous CD8$^+$ T cells are required for protective antitumor memory in this setting. Our findings thus demonstrate the stimulatory effect of CD40L-overexpressing CAR T cells on innate and adaptive immune cells, and provide a rationale for using CD40L-overexpressing CAR T cells to improve immunotherapy responses.

[1] Louis V. Gerstner Jr. Graduate School of Biomedical Sciences, Memorial Sloan Kettering Cancer Center, New York, NY, USA. [2] Department of Medicine, Memorial Sloan Kettering Cancer Center, New York, NY, USA. [3] Present address: Department of Pathology, University of California San Francisco, San Francisco, CA, USA. ✉email: brentjer@mskcc.org

Chimeric antigen receptors (CAR) are synthetic fusion proteins with an extracellular antigen-recognition domain and intracellular T cell stimulation domain, allowing T cell-mediated targeting of surface molecules on tumor cells independently of peptide-major histocompatibility complex (pMHC) presentation[1]. CD19-targeted CAR T cells have shown impressive results in patients with relapsed or refractory B cell malignancies, leading to the approval of anti-CD19 CAR T cell therapy in patients with diffuse large B cell lymphoma and pediatric B-acute lymphocytic leukemia[2,3]. Whereas complete remission after anti-CD19 CAR T cell infusion can be reached in a large fraction of patients, a significant portion of responding patients relapses with CD19-negative disease and others do not respond at all[2,4]. This observation warrants improved T cell-based immunotherapies to target and eliminate malignant tumor cells.

Most T cell-based immunotherapies are focused on directly increasing the number of tumor-targeted T cells by adoptively transferring T cells into patients, removing immune-inhibitory checkpoints that act on the endogenous repertoire of the polyclonal T cell population, redirecting the endogenous T cell population to the tumor via bispecific antibodies, and/or genetically engineered T cells with CARs that allow supraphysiological antitumor T cell responses[5]. All these strategies directly manipulate and redirect immune responses on the T cell level. We have shown that CD40L-overexpressing CAR T cells have an increased antitumor effect, license antigen-presenting cells (APCs) in vivo, activate endogenous T cells, and protect mice from CAR-antigen negative tumor challenge[6]. This strategy is an example of mobilizing immune effectors besides the adoptively transferred CAR T cells to eradicate tumor cells. However, the immune subpopulations necessary for relaying the information from CD40L-overexpressing CAR T cell to endogenous T cells and protecting mice from CAR-antigen negative tumor outgrowth remain to be determined.

Conventional DCs (cDCs), as opposed to plasmacytoid DCs, are the most potent APCs and can be further subdivided into cDC1 and cDC2 populations[7]. cDCs express high levels of MHC-II and CD11c in both humans and mice. The transcription factors BATF3, IRF8, and ID2 are essential for cDC1 development, whereas cDC2s depend on the transcription factors RELB, IRF4, and ZEB[8]. cDC2s are predominantly involved in initiating CD4+ T cell responses against nematodes and viral infections[9,10]. So far, there is limited understanding of cDC2 function in the immune antitumor response, but a recent study has identified cDC2s in mice and humans and their involvement in CD4+ T cell activation[11].

cDC1s express surface CD8α and CD103 (integrin αE) in lymphoid and non-lymphoid tissue, respectively. Both lymphoid and non-lymphoid tissue cDC1s share a very similar transcriptional profile and a central role in the adaptive immune response by cross-presenting antigen to cytotoxic CD8+ T cells in antiviral and antitumor responses[7,12,13]. In humans, cDC1s are identified by CD141/BDCA3 surface expression[14] and seem to be excluded from tumor tissue compared to matched, healthy tissue[15]. Their role in tumor rejection is further supported by the finding that high levels of intratumoral BDCA3+ cDC1s correlate with responsiveness to anti-PD-1 immunotherapy in melanoma patients[16]. Thus, several preclinical tumor transplantation studies aimed at increasing the accumulation of tumor-resident cDC1s and noted that NK cell-derived fms like tyrosine kinase 3 ligand (FLT3L) and other cDC1 chemoattractants stimulated cDC1 recruitment to the tumor and controlled further tumor growth[16,17]. Importantly, the accumulation of tumor-resident cDC1s improved CD8+ T cell expansion and responses to anti-PD-L1 treatment[18].

T cell-mediated antitumor responses are well characterized and can be based on recognition of non-mutated cancer-related antigens or neoantigens derived from mutated proteins[19,20]. The antigen is recognized by the TCR of the CD4+ or CD8+ T cell on presented MHC-II or MHC-I, respectively. The importance of CD8+ T cell-mediated tumor control through pMHC-I:TCR interactions is highlighted by the observation that MHC-I or β2M loss in tumor cells—both resulting in the absence of antigen presentation on the cancer cells surface—leads to tumor immune evasions and subsequent tumor outgrowth in patients[21,22]. Still, long-term survival for up to 10 years has now been described in patients with metastatic disease who were treated with T cell-mobilizing immunotherapies[23]. Similar results have been described for B-ALL patients treated with anti-CD19 CAR T cells in a long-term follow-up[4].

However, many patients relapse with CAR-antigen-negative disease at later time points. Thus, a sustained antitumor response that is based on the highly cytotoxic effector function of the CAR T cell, plus the recruitment of cytotoxic non-CAR T cells recognizing tumor cell-specific antigens is explored in this study. We demonstrate that m1928z-CD40L CAR T cells can induce long-lived immune cell-based antitumor memory and, thereby, provide protection from CAR-antigen-negative tumor outgrowth in a preclinical setting.

## Results

**m1928z-CD40L CAR T cells upregulate CCR7 on tumor-resident cDCs and skew the intratumoral DC population towards the CD11b−CD103− DN and CD11b−CD103+ cDC1 phenotype.** We have previously described in vivo licensing of APCs through CD40L-overexpressing CAR T cells in lymphoid tissue, but not in tumor tissue, which was most prominent 7 days after adoptive cell transfer (ACT)[6]. This prompted us to analyze earlier time points after ACT by quantifying DC recruitment to both tumor and spleen (Fig. 1a). m1928z-CD40L CAR T cell treatment did not increase the recruitment of bulk MHC-II+CD11c+ DCs to the tumor tissue over time (Fig. 1b). Compared to m1928z CAR T cell treatment, CD40L-overexpressing CAR T cells induced the accumulation of splenic MHC-II+CD11c+ DCs by day 7 after ACT (Fig. 1c). Besides the spleen, the secondary lymphoid organs are also comprised of lymph nodes where immune cell interactions are organized between members of the innate and adaptive immune system. MHC-IIhiCD11cint migratory DCs (migDC) transport antigen from surrounding tissue to draining lymph nodes where they present antigen to circulating T cells for activation[13,24,25]. m1928z-CD40L CAR T cell treatment increased the migDC population in tumor-draining lymph nodes (tdLNs) (Fig. 1d). This increase in migDCs was instructed via CD40/CD40L interactions, as Cd40−/− mice lacked an increase in migDCs after m1928z-CD40L CAR T cell treatment (Fig. 1d). By analyzing these different anatomical sites, we noticed that m1928z-CD40L CAR T cell treatment has no effect on DC tumor infiltration numbers—neither at earlier, nor later time points—nor does it affect the lymphoid compartment until one week after ACT.

Conventional DCs can be further divided into cDC1 and cDC2 subpopulations. Both subpopulations have been described to have roles in antitumor immune responses, where they control T cell immunity[18,26]. Thus, we investigated DC subpopulations in m1928z-CD40L CAR T cell-treated mice. cDC1 and cDC2 populations can be immunophenotyped based on surface marker expression. Identified by high expression of the conventional DC markers MHC-II and CD11c, cDC1 populations in non-lymphoid tissue express the integrin CD103 and are CD11b−, whereas cDC2s are CD11b+CD103− (Supplementary Fig. 1A). In the lymphoid tissue, the cDC1 population loses its CD103 expression and instead is identified by CD8α expression, with cDC2 cells maintaining their

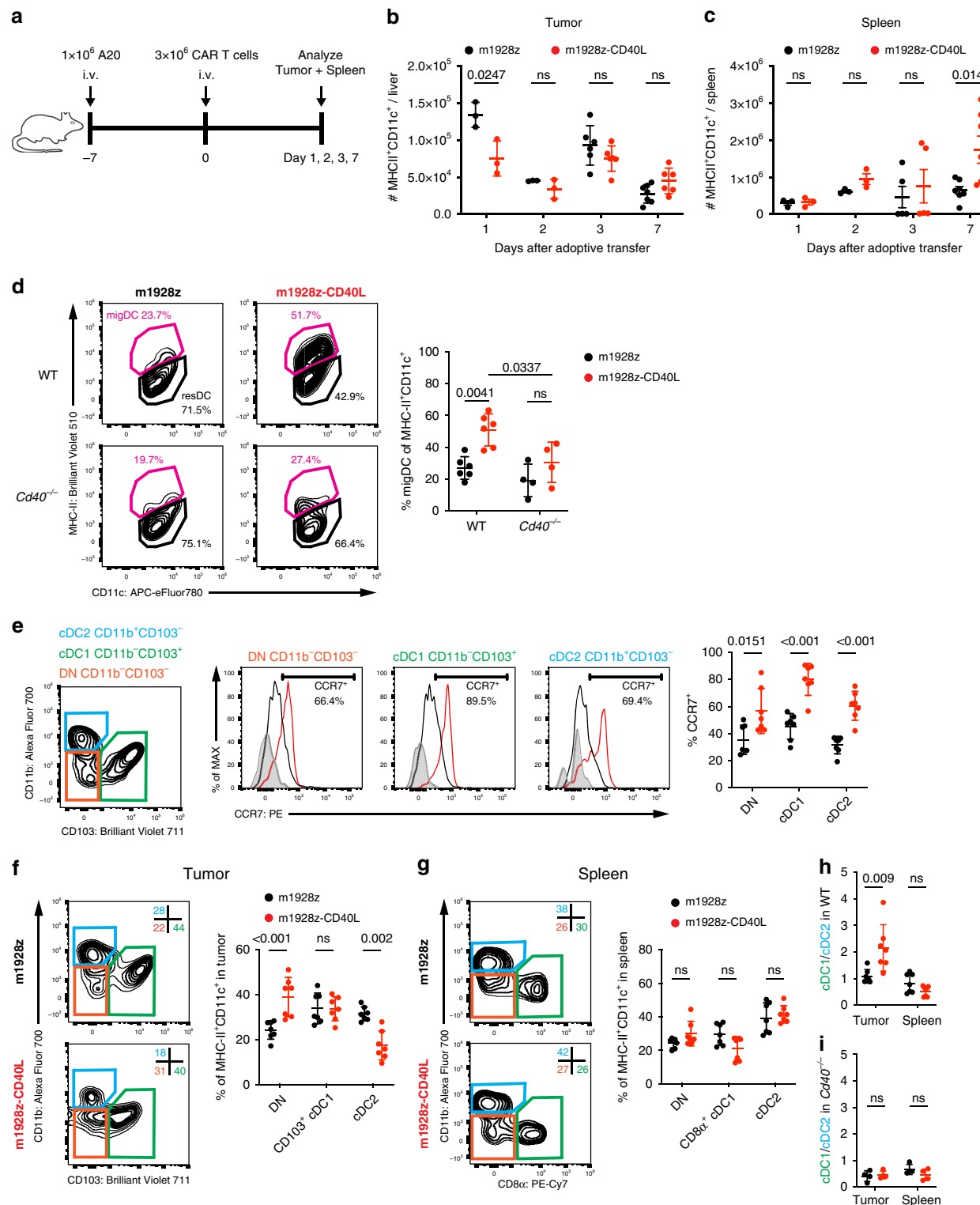

CD11b+ CD8α+ status (Supplementary Fig. 1B). Recently, CCR7-expressing CD103+ DCs were identified as trafficking between tumor site and tdLN, transporting tumor antigen, and priming a T cell anti-tumor response[25]. The chemokine receptor CCR7 directs DC trafficking to LNs[27] and we observed increased CCR7 expression on all three tumor-resident MHC-II+CD11c+ DC populations after m1928z-CD40L CAR T cell treatment: CD11b− CD103− DNs, CD11b−CD103+ cDC1s, and CD11b+CD103− cDC2s (Fig. 1e). This suggested that m1928z-CD40L CAR T cell treatment recruits DCs to the spleen and tdLNs by inducing upregulation of CCR7 on tumor-resident DCs.

Besides upregulation of CCR7 surface-level expression, we wanted to investigate if m1928z-CD40L CAR T cell treatment affects the relative composition of the three intratumoral cDC subpopulations. CD40L-CAR T cell treatment increased the DN population, maintained the cDC1 population, resulting in a relative decrease in the cDC2 fraction (Fig. 1f and Supplementary Fig. 1C). These changes were not noticeable in the splenic and

**Fig. 1 m1928z-CD40L CAR T cells upregulate CCR7 on tumor-resident dendritic cells (DCs) and skew the intratumoral DC population towards the CD11b-CD103- double-negative (DN) and CD11b-CD103+ cDC1 phenotype. a** Experimental layout for (**b, c**). **b, c** Absolute number of MHC-II+CD11c+ DCs (CD45+Gr1−CD19−CD3e− pre-gates) in tumor (**b**) and spleen (**b**) of A20.GL tumor-bearing mice treated as outlined in (**a**). Each dot represents one mouse (Day 1, 2: $n = 3$/group; Day 3: $n = 6$/group; Day 7: $n = 6$-7/group). Data are plotted as mean ± SD. *p*-values were obtained from an unpaired two-tailed Student's *t* test. **d** A20.GL tumor-bearing wild-type (WT) or $Cd40^{-/-}$ mice received $3 \times 10^6$ CAR T cells intravenously (i.v.). The percentage of MHC-IIhiCD11cint migratory DCs (migDC) in tumor-draining lymph-nodes (tdLNs) was analyzed on day 7. Data are plotted as mean ± SD and pooled from two independent experiments (WT: $n = 6$/group; $Cd40^{-/-}$: $n = 4$/group). *p*-values were determined by two-way ANOVA test. **e** A20.GL tumor-bearing mice received $3 \times 10^6$ CAR T cells i.v. and CCR7 surface expression was analyzed on day 7 on CD11b−CD103− double-negative (DN) (orange), CD11b−CD103+ cDC1 (green), and CD11b+CD103+ cDC2 (blue) populations in the tumor. Gray histogram, flow minus one. **f** A20.GL tumor-bearing mice received $3 \times 10^6$ CAR T cells i.v. and the percentage of CD11b−CD103− DN (orange), CD11b−CD103+ cDC1 (green), and CD11b+CD103+ cDC2 (blue) populations in the tumor was analyzed on day 7. **g** A20.GL tumor-bearing mice received $3 \times 10^6$ CAR T cells i.v. and the percentage of CD11b−CD8α− DN (orange), CD11b−CD8α+ cDC1 (green), and CD11b+CD8α− cDC2 (blue) populations in the spleen was analyzed on day 7. **h** The cDC1/cDC2 ratio in A20.GL tumor-bearing WT mice is plotted in the tumor and spleen of mice treated in (**f, g**). **i** The cDC1/cDC2 ratio in A20.GL tumor-bearing $Cd40^{-/-}$ mice is plotted on day 7 after receiving $3 \times 10^6$ CAR T cells. Data in (**e-i**) is plotted as mean ± SD and pooled from two independent experiments. Each dot represents one mouse (**e-h**, $n = 7$/group; **i**, $n = 4$/group). *p*-values were obtained from an unpaired two-tailed Student's *t* test. ns, non-significant; i.v. intravenous; resDC, resident DC. Source data are provided as a Source Data file.

tdLN cDC populations (Fig. 1g and Supplementary Fig. 1D–F). These findings documented a change in the dendritic cell compartment of the tumor tissue after m1928z-CD40L CAR T cell treatment, wherein m1928z-CD40L CAR T cells skew the tumor-resident cDC1/cDC2 ratio in favor of the cDC1 population (Fig. 1h and Supplementary Fig. 1C). Lack of cDC1/cDC2 ratio increase in $Cd40^{-/-}$ mice showed that m1928z-CD40L CAR T cells mediate this effect in vivo via CD40/CD40L interactions (Fig. 1i). These observations highlighted the effect m1928z-CD40L CAR T cells have on intratumoral DC subpopulations and prompted the question of its relevance in the antitumor response.

**Improved antitumor response of m1928z-CD40L CAR T cells requires presence of Batf3-expressing cDC1.** Next, we wanted to prevent cDC1 accumulation in the tumor and, thereby, assess its necessity in the m1928z-CD40L CAR T cell-mediated antitumor response. The transcription factor BATF3 is important for the development of cDC1s, as mice lacking *Batf3* expression are deficient in CD8α+ DCs and tumor-resident CD103+ DCs, making them more susceptible to CD8+ T cell-controlled viral infections and tumor growth[12,26]. We challenged both wild-type and $Batf3^{-/-}$ mice with GFP+ luciferase-expressing A20 lymphoma cells (A20.GL) and, as expected, no CD11b−CD103+ cDC1 cells were present in the tumor of $Batf3^{-/-}$ mice (Fig. 2a). As we have previously reported[6], m1928z-CD40L CAR T cells increase survival of A20 tumor-bearing wild-type mice without the need for prior lymphodepletion (Fig. 2b). Subsequent treatment with CAR T cells showed that m1928z-CD40L CAR T cells in $Batf3^{-/-}$ mice improves survival of tumor-bearing mice (Fig. 2b), demonstrating that the cDC1 population is not solely responsible for in vivo m1928z-CD40L CAR T cell function. However, their presence in wild-type mice significantly improved survival and allowed complete tumor clearance by m1928z-CD40L CAR T cells in 40% of mice (Fig. 2b). Taken together, for m1928z-CD40L CAR T cells to exert their full antitumor response, the presence of cDC1s is required.

The specific depletion of cDC2s remains challenging, because so far no marker has been identified that is exclusively expressed in cDC2s[28]. Use of CD11c-driven Cre recombinase expression to conditionally delete a cDC2 transcription factor is not restricted to the DC compartment, as CD11c is also expressed in macrophages and lymphoid cells[29,30]. This warrants further development of new mouse models that specifically deplete cDC2s, in order to gain a better understanding of their function in our system and other immune contexts.

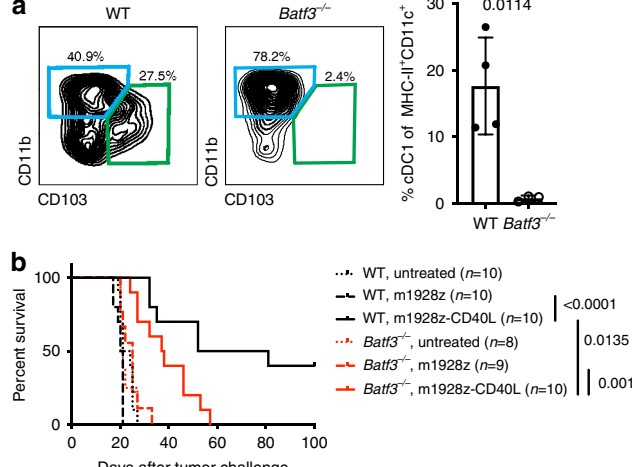

**Fig. 2 Improved antitumor response of m1928z-CD40L CAR T cells requires presence of Batf3-expressing cDC1. a** Flow cytometry contour plots of the MHC-II+CD11c+ DC population in tumors of WT and $Batf3^{-/-}$ mice on day 14 after A20.GL intravenous injection. Gates highlight CD11b+CD103− cDC2 (blue) and CD11b−CD103+ cDC1 (green) cells. Percentage of CD11b−CD103+ cDC1s in WT and $Batf3^{-/-}$ mice is plotted on the right. Data are plotted as mean ± SD. Each dot represents one mouse (WT: $n = 4$; $Batf3^{-/-}$, $n = 3$). *p*-value was determined using an unpaired two-tailed Student's *t* test. **b** Survival of BALB/c WT or $Batf3^{-/-}$ mice challenged with $1 \times 10^6$ A20.GL cells and treated with $3 \times 10^6$ CAR T cells on day 7. *p*-values were determined by a two-tailed log-rank (Mantel-Cox) test. The summary of two independent experiments is plotted. Source data are provided as a Source Data file.

**m1928z-CD40L CAR T cells stimulate tumor-resident CD11b−CD103− DN cDCs to proliferate, up regulate IRF8, and differentiate to cDC1s.** Next we wanted to investigate how the different CAR T cell treatments affect the cDC subpopulations as seen in Fig. 1. cDC1s consistently expressed the highest levels of the co-stimulatory molecule CD86 in all tissues analyzed (spleen, tumor, tdLN; Supplementary Fig. 2). m1928z-CD40L CAR T cell treatment did not further increase the high CD86 surface levels on cDC1s, whereas the lower CD86 expression on splenic DN and cDC2 cells was elevated following treatment (Supplementary Fig. 2B). This licensing of splenic cDCs matched the increase of splenic DC numbers seen in Fig. 1c, whereas tumor-resident cDCs remained unchanged (Fig. 1b and Supplementary Fig. 2C),

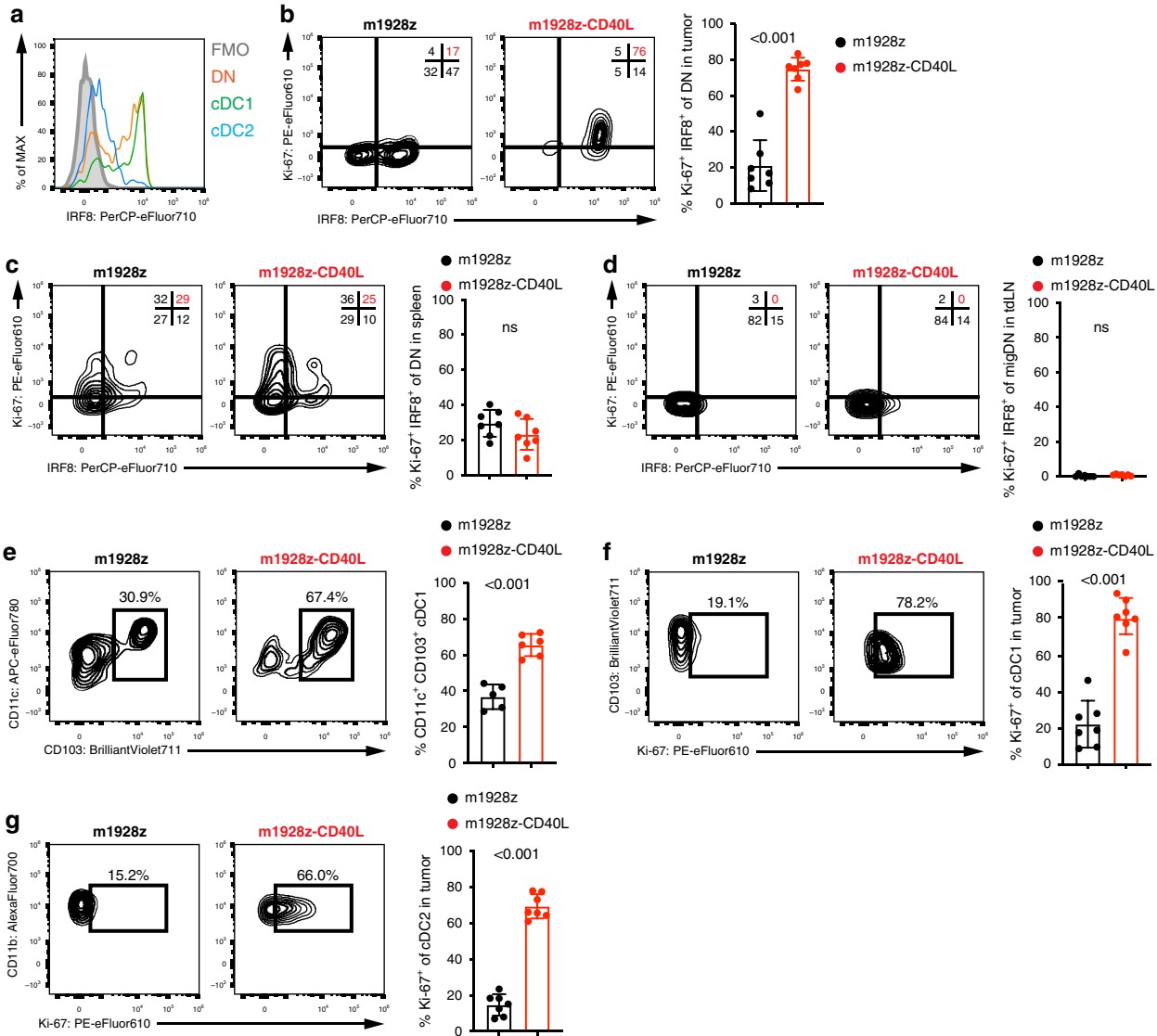

**Fig. 3 m1928z-CD40L CAR T cells stimulate tumor-resident CD11b-CD103- DN cDCs to proliferate, upregulate IRF8, and differentiate to cDC1s. a** IRF8 expression in CD11b−CD103− double-negative (DN) (orange), CD11b−CD103+ cDC1 (green), and CD11b+CD103+ cDC2 (blue) in the tumor of untreated A20.GL tumor-bearing mice. FMO, flow minus one. **b–d** Ki-67 and IRF8 expression shown as flow cytometry contour plots in CD11b−CD103− DN cDCs in the tumor (**b**), spleen (**c**), and tumor-draining lymph nodes (tdLN) (**d**) of A20.GL tumor-bearing mice on day 7 after CAR T cell treatment. Percentage of Ki-67+IRF8+ DN cDCs is summarized from two independent experiments ($n = 7$/group). **e** CD45.2+ A20.GL tumor-bearing mice were treated with $3 \times 10^6$ CAR T cells i.v. and CD45.2+ CD11b−CD103− DN cDCs were isolated from the tumor on day 3 by FACS. Sorted CD45.2+ DN cells were cultured in vitro on a CD45.1+ bone-marrow stromal layer for 3 days and the percentage of CD11c+CD103+ cDC1s of all CD45.2+ cells was analyzed. Shown are representative contour plots and the quantification of the percentage of CD11c+CD103+ cDC1s. Each dot represents one in vitro culture. Data were collected from two independently performed experiments (m1928z, $n = 5$; m1928z-CD40L, $n = 6$). **f, g** Ki-67 expression shown as contour plots in CD11b−CD103+ cDC1s (**f**) and CD11b+CD103− cDC2s (**g**) in the tumor of A20.GL tumor-bearing mice on day 7 after CAR T cell treatment. Percentage of Ki-67+ cells is summarized from two independent experiments ($n = 7$/group). Data in (**b–g**) is plotted as mean ± SD. p-values were obtained from an unpaired two-tailed Student's t test. ns, non-significant. Source data are provided as a Source Data file.

suggesting that cDCs respond differently to m1928z-CD40L CAR T cell treatment depending on their tissue site.

Focusing on the peripheral, differentiated cDC populations, we next assessed protein expression of the IRF8 transcription factor in tumor-derived cDC populations (Fig. 3a). In the periphery, IRF8 controls survival and function of terminally differentiated cDC1s[31,32]. Furthermore, increased IRF8 expression in CD11b−CD103− DN cells was shown to promote their differentiation into mature CD103+ cDC1s[18]. Thus, we hypothesized that CD40L-CAR T cell treatment skews the cDC1/cDC2 ratio towards the cDC1 populations by stimulating the DN cells to expand,

upregulate IRF8, and differentiate into cDC1s. We specifically noticed upregulation of IRF8 (readout of DN-to-cDC1 differentiation) and Ki-67 (readout for proliferation) in DN cells treated with CD40L-CAR T cell-treated mice (Fig. 3b). The increased expression of Ki-67 in the tumor-derived DN cells also correlated with the observed increase of the DN population in the tumor of CD40L-CAR T cell-treated mice (Fig. 1f), indicating that DN cells receive a proliferative signal upon CD40L-CAR T cell treatment. Intriguingly, splenic DN cells and DN cells from the tdLNs did not upregulate Ki-67 or IRF8 (Fig. 3c, d). Overall, tdLN cDCs did not present with a stimulated phenotype, as

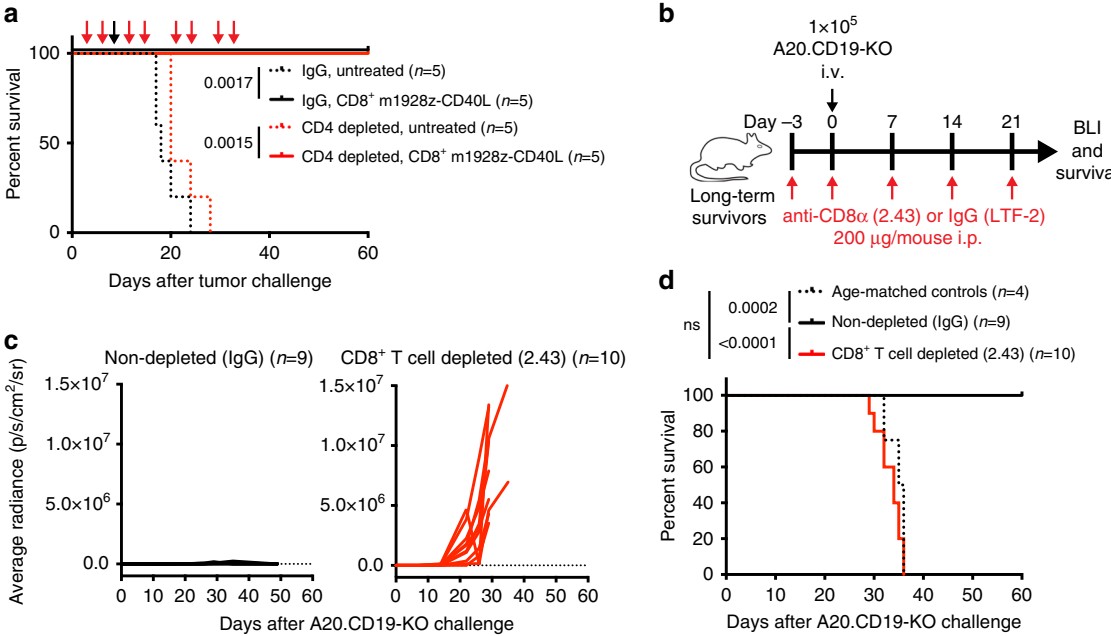

**Fig. 4 CD8+ T cells are necessary for m1928z-CD40L CAR T cell-mediated protection against antigen-negative tumor growth. a** Survival of naïve BALB/c mice injected with $1 \times 10^6$ A20.GL cells intravenously (i.v.) and either left untreated or treated with $3 \times 10^6$ CD8+ m1928z-CD40L CAR T cells i.v. on day 7 (black arrow). For CD4+ T cell depletion, two cohorts of mice received 200 μg of anti-CD4 depletion antibody (GK1.5) by intraperitoneal (i.p.) injection 2x per week for 3–4 weeks (red arrows). One of two representative experiments is shown. **b** Experimental scheme for (**c**, **d**). **c** Tumor burden of mice injected with luciferase-expressing CD19neg A20.CD19-KO cells was monitored using bioluminescence imaging. Average radiance per whole animal is plotted for the IgG treated mice ($n = 9$) and the CD8+ T cell-depleted mice ($n = 10$). **d** Survival of mice treated in (**b**, **c**). Naïve age-matched BALB/c mice were used as controls. All p-values in figure were determined by a two-tailed log-rank (Mantel-Cox) test. ns, non-significant. Source data are provided as a Source Data file.

suggested by the lack of CD86 or Ki-67 upregulation (Supplementary Fig. 2D–G). This suggested a tumor-specific effect of cDC DN stimulation after m1928z-CD40L treatment.

Next, we wanted to assess if IRF8 upregulation in the DN population leads to DN-to-cDC1 differentiation. To address this, DN cells were isolated by fluorescence-activated cell sorting (FACS) from tumors of m1928z and m1928z-CD40L CAR T cell-treated mice and cultured ex vivo for 3 days to assess their potential to differentiate to cDC1s without any further stimuli. Both, DN cells from m1928z and m1928z-CD40L CAR T cell-treated mice differentiated into cDC1s ex vivo, albeit DN cells from m1928z-CD40L CAR T cell-treated mice differentiated into cDC1s at a significantly higher rate compared to m1928z CAR T cell-treated mice (Fig. 3e). Importantly, cDC1 and cDC2 cultured cells maintained their CD103+ and CD103− phenotype, respectively, regardless of prior stimulation (Supplementary Fig. 3A, B). Together, this implies that CD40L-CAR T cells affect the intratumoral cDC1/cDC2 ratio by stimulating CD11b−CD103− DN cell proliferation, upregulation of the cDC1-skewing IRF8 transcription factor, and, consequently, differentiation of DN cDCs to cDC1s in the tumor tissue.

Besides DN-to-cDC1 differentiation, m1928z-CD40L CAR T cell treatment also promoted the proliferation of cDC1s and cDC2s, as suggested by Ki-67 staining (Fig. 3f, g). Curiously, the increased proliferation in all tumor-resident cDC subsets did not result in an increase in absolute cDC numbers after CD40L-CAR T cell treatment (Fig. 1b), suggesting that the proliferating tumor-resident cDCs migrate out of the tissue to lymphoid structures, where they are present at higher numbers (Fig. 1c, e).

**CD8+ T cells are necessary for m1928z-CD40L CAR T cell-mediated protection against antigen-negative tumor growth.** With the increased priming of tumor-infiltrating CAR+ and

CAR− T cells[6] and an impaired antitumor response in m1928z-CD40L CAR T cell-treated Batf3−/− mice (Fig. 2d), we hypothesized that m1928z-CD40L CAR T cells prime the tumor-infiltrating T cell population through the cDC1 population. Due to the known function of cDC1 cells cross-presenting antigen to CD8+ T cells, we focused on analyzing the CD8+ T cell compartment. As expected, after A20.GL tumor challenge and CAR T cell treatment, CARneg CD8+ T cells in m1928z-CD40L CAR T cell-treated mice produced more IFNγ after ex vivo phorbol 12-myristate 13-acetate (PMA)/ionomycin stimulation (Supplementary Fig. 4A, B). However, this elevated IFNγ production was sustained in the absence of cDC1 cells in Batf3−/− mice (Supplementary Fig. 4A, B). This indicated that m1928z-CD40L CAR T cells provide a permissive environment in vivo that allows endogenous CD8+ T cells to robustly produce IFNγ independently of cDC1 cells.

To assess if the absence of cDC1 cells has any effect on the adoptively transferred T cell population, which might explain the impaired antitumor response in Batf3−/− mice, IFNγ production in CD3+CAR+ T cells was assessed. Again, more IFNγ was detected in m1928z-CD40L CAR T cells compared to m1928z CAR T cells and this difference was not affected by genetic deletion of cDC1 cells in Batf3−/− mice (Supplementary Fig. 4C, D). This suggested that cDC1 cells are not responsible for the increased effector cytokine production observed in both CARneg and CAR+ T cells of m1928z-CD40L CAR T cell-treated mice.

Assessment of cytokine production was done by ex vivo intracellular cytokine staining after non-specific activation of T cells by the diacylglycerol analog PMA and the calcium ionophore ionomycin. Together, these stimulants lead to protein kinase C activation (via PMA) and calcium release (via ionomycin) in T cells and activate T cells downstream of TCR-induced activation. Thus, analysis of PMA/ionomycin stimulated

cell populations provides a readout for general cell activation potential, but not a readout for functional cell-specific tumor recognition.

Finally, we wanted to identify the cell population in the cured mice that mediate the protection against CAR-antigen-negative tumor outgrowth. To explore the possibility that endogenous CD4+ T cells are necessary for the improved antitumor response of m1928z-CD40L CAR T cell treatment, we took advantage of the finding that CD8+ CAR T cells alone, but not CD4+ CAR T cells, could cure A20.GL tumor-bearing mice (Supplementary Fig. 5A, B). This allowed isolated antibody-mediated depletion of CD4+ T cells in the context of CD8+ m1928z-CD40L CAR T cell treatment, to assess the role of non-CAR CD4+ T cells in the antitumor response. CD4+ T cells were depleted with the anti-CD4 antibody clone GK1.5 in A20.GL tumor-bearing mice before and after CD8+ m1928z-CD40L CAR T cell treatment (Fig. 4a). CD4+ T cell depletion was confirmed by flow cytometry in the peripheral blood of GK1.5-treated mice with a different anti-CD4 antibody clone (RM4-5) on the day of ACT (day 7) and a later time point (day 21) to make sure that no CD4+ T cells are present that could potentially aid during the initial antitumor CAR T cell response (Supplementary Fig. 5C). Survival of CD4-depleted mice demonstrated that CD4+ T cells are not necessary for the improved antitumor response through m1928z-CD40L CAR T cell treatment (Fig. 4a).

Focusing on the CD8+ T cell compartment, antibody-mediated depletion of CD8+ T cells before and/or after ACT would also lead to depletion of CD8+ T cells in the CAR T cell product, due to the long half-life of the depletion antibody and its systemic persistence. This would make it impossible to attribute the observed results to either the endogenous CD8+ T cell population or the adoptively transferred CAR T cells. To circumvent this problem, we decided to investigate the contribution of CD8+ T cells to the memory response against CAR-antigen-negative tumor cell challenge. Long-term surviving mice that were initially cured from A20.GL tumor challenge by m1928z-CD40L CAR T cells were injected with CAR-antigen-negative A20.CD19-KO cells to exclude any CAR T cell-mediated antitumor response by persisting CAR T cells (Fig. 4b). Nineteen mice that were tumor free by bioluminescent imaging at day 50+ after initial luciferase-expressing A20.GL tumor challenge and m1928z-CD40L CAR T cell treatment were collected from three different previous experiments and separated into two cohorts (Supplementary Fig. 5D, E, F). Ten of 19 mice were CD8+ T cell-depleted by intraperitoneal injection with anti-CD8 antibody clone 2.43 (Fig. 4b). The remaining nine mice received the IgG control antibody. Complete CD8+ T cell depletion was confirmed (Supplementary Fig. 5G) and growth of luciferase-expressing A20.CD19-KO tumor cells were measured over time (Fig. 4c). Mice cured from primary A20.GL tumor challenge that were depleted of CD8+ T cells were not able to control A20.CD19-KO tumor outgrowth, unlike the IgG control mice (Fig. 4c). This resulted in a lack of survival due to disease progression (Fig. 4d). All relapsed mice died from outgrowth of CD19− tumor cells, indicating that CD8+ T cell depletion did not cause reemergence of residual CD19+ tumor cells from the first tumor challenge (Supplementary Fig. 5H, I).

## Discussion

This study describes the recruitment of tumor-specific endogenous CD8+ T cells after m1928z-CD40L CAR T cell treatment. Their mobilization was dependent on the presence of cross-presenting cDC1s and their elimination via anti-CD8 antibody-mediated cell depletion made mice susceptible to CAR-antigen-negative tumor cell outgrowth. These findings highlight the induction of a sustained host antitumor response by m1928z-CD40L CAR T cells.

This effect is presumably mediated by the recognition of tumor-associated antigens (TAAs) presented on MHC-I by the cancer cells to the cytotoxic CD8+ T cells, as described in mice and human cancer patients[33,34]. The recognition of these TAAs by host T cells is especially important under the aspect of tumor heterogeneity and outgrowth of CAR-antigen-negative tumor cells after ACT, as seen in the clinic[4]. This early CD8+ T cell priming via CD40L-overexpressing CAR T cells seemed to generate a long-lived memory response that protected mice from CAR-antigen-negative tumor outgrowth. More careful analysis of TAA presentation on A20 tumor cells is required to assess the extent of immunodominant epitope recognition by the CD8+ T cell population. Exome sequencing of tumor cells in combination with in silico prediction of epitope presentation could identify such immunodominant epitopes and potential CD8+ T cell clones recognizing them[35]. This would inform the extent of CD8+ T cell tumor recognition and allow differentiation between a dominant clonal response and a possible oligoclonal T cell antitumor response.

Since A20 tumor cells are derived from transformed B cells, they express high levels of MHC-II, making them susceptible to CD4+ T cell-mediated recognition. However, the CD4+ T cell-specific antitumor response did not seem to be relevant, as prior depletion of CD4+ T cells did not lessen the m1928z-CD40L CAR T cell-mediated antitumor response. CD4+ T cells are important in helping to initiate a cytotoxic CD8+ T cell response through CD40 signaling on APCs via surface CD40L expression[36]. Overexpression of CD40L on CAR T cells—both on CD4+ and CD8+ CAR T cells—obviates the need for CD4+ T cell help, similar to earlier studies demonstrating efficient CD8+ T cell priming with agonistic anti-CD40 antibodies independently of CD4+ T cell help[37,38]. Thus, CD4+ T cells are dispensable in mounting an efficient antitumor response, as long as CD40 signaling on APCs in provided by an alternative source, such as CD40L-overexpressing CAR T cells.

Cytokine production of endogenous CD8+ T cells was evident in Batf3−/− mice, despite the absence of cross-presenting cDC1s. Cross-priming of CD8 T cells independently of Batf3-expressing cDC1s has been described. CD169+ macrophages have been identified as possible antigen cross-presenters for CD8 T cell stimulation in LNs[39,40]. In our system, we have previously reported the activation of both macrophages and DCs[6], warranting further work to establish a potential stimulatory role of macrophages in CAR T cell-treated mice. Whereas we show that lack of Batf3-expressing cDC1s impairs the m1928z-CD40L CAR T cell antitumor response, identification and depletion of other cross-presenting cells could possibly completely ablate the antitumor response and provide evidence that other non-cDC1s are involved as well.

Why the cDC1/cDC2 ratio increases in tumors of m1928z-CD40L CAR T cell-treated mice is unclear and warrants further investigation. The accumulation of cDC1s in the tumor tissue has been attributed to several NK cell-derived cytokines such as CCL5, FLT3L, and XCL1[16,17]. Conventional DCs in peripheral tissue have a half-life of about 3–6 days and are maintained by tissue-resident pre-cDCs that originate in and exit from the bone marrow[41,42]. This process can be observed in a mouse model of influenza infection, when pre-cDCs traffic to the infected lung tissue and locally increase the cDC numbers[43]. We see increased proliferation of cDCs after CD40L-CAR T cell treatment only in the tumor and not in lymphoid tissue. More importantly, CD40L-CAR T cell treatment skews the cDC1/cDC2 ratio in favor of the cDC1s by promoting differentiation of progenitor IRF8+ DN progenitor cells to cDC1s. This is similar to published results,

were homeostasis and generation of cDCs in peripheral tissue is maintained by mobilization of progenitor cDCs from the bone marrow[18,43]. Both endogenous and exogenously applied FLT3L are instructive in mediating this effect[18,41,42,44], suggesting a pathway that potentially is activated upon CD40L-CAR T cell treatment. It is unclear if pre-cDCs found in different tissues respond to differentiation signals differently, warranting further analysis of progenitor DCs residing in different tissues.

The priming of CD8$^+$ T cells by cDC1s has been described to happen in the tdLNs after CD103$^+$ cDC1s take up the tumor antigen, upregulate CCR7 on their surface to home to the lymph node, where they then cross-present antigen to LN-resident CD8$^+$ T cells[25]. We noticed efficient priming of splenic CD8$^+$ T cells, indicating a more systemic, rather than local, CD8$^+$ T cell activation by m1928z-CD40L CAR T cell treatment. This could potentially be explained by the accumulation of CAR T cells in the spleen, a site of high anti-CD19 CAR-antigen concentration. There, they induce licensing of APCs[6] and provide a permissive environment for T cell priming. Our use of a disseminated lymphoma model has distinct characteristics compared to bona fide solid tumor models, with differences in stromal involvement and an immunosuppressive tumor microenvironment[45,46]. Use of a different tumor model, not targeting a ubiquitous antigen such as CD19 on B cells, could help to delineate if CD40L-modifed CAR T cells can prime CD8$^+$ T cells via APC licensing at a local level. However, m1928z-CD40L CAR T cell treatment did specifically lead to an increase in the tumor-resident CD103$^+$ cDC1 population, whereas lymphoid CD8$\alpha^+$ cDC1s were not elevated. This suggests priming of CD8$^+$ T cells in the tumor microenvironment, which is possible and has been described for TILs in a previous study[47]. More detailed analysis of tumor-infiltrating versus tdLN-resident CD8$^+$ T cells after m1928z-CD40L CAR T cell treatment could help to delineate the logistics of cDC1-CD8$^+$ T cell interactions. In addition, it could inform us in which tissue environment the priming happens. This would help to design additional strategies improving this priming process, which is crucial for mounting a sustained, endogenous antitumor response.

## Methods

**Animal models**. All mice were bred and co-housed under SPF conditions in the animal facility of Memorial Sloan Kettering Cancer Center. All experiments were performed in ethical accordance with and upon approval by the MSKCC Institutional Animal Care and Use Committee (MSKCC protocol #00-05-065). Wild-type BALB/c mice were purchased from Charles River. BALB/c CD45.1 (CByJ.SJL(B6)-*Ptprc$^a$*/J) were purchased from Jackson laboratories. BALB/c *Cd40$^{-/-}$* (CNCr.129P2-*Cd40$^{tm1Kik}$*/J) were kindly provided by Dr. Anna Valujskikh and bred in-house. BALB/c *Batf3$^{-/-}$* (C.129S-*Batf3$^{tm1Knm}$*/J) were kindly provided by Dr. Barney Graham and bred in-house. 8–12-week old female mice were used in all experiments, unless indicated differently. Mice challenged with firefly luciferase-expressing tumor were imaged via bioluminescence to confirm equal tumor load and randomized to different treatment groups one day before treatment. Mice were euthanized via CO$_2$ inhalation when tumor growth led to a weight gain of 20% due to a distended abdomen or when mice suffered from hind limb paralysis. The investigator was blinded when assessing the outcome.

**Cell lines**. A20 cells (catalog number TIB-208) and Phoenix-ECO packaging cells (catalog number CRL-3214) were purchased from ATCC. All cell lines and culture experiments were maintained in RPMI-1640 supplemented with 10% heat-inactivated FBS, nonessential amino acids, 1 mM sodium pyruvate, 10 mM HEPES, 2 mM L-glutamine, 1% penicillin/streptomycin, 11 mM glucose, and 2 μM 2-mercaptoethanol. Cell lines were routinely tested for potential mycoplasma contamination.

**Generation of retroviral constructs**. Plasmids encoding the CAR construct in the SFG γ-retroviral vector[48] were used to transfect gpg29 fibroblasts (H29) with the ProFection Mammalian Transfection System (Promega) according to the manufacturer's instructions in order to generate vesicular stomatitis virus G-glycoprotein-pseudotyped retroviral supernatants. These retroviral supernatants were used to construct stable Moloney murine leukemia virus-pseudotyped retroviral particle-producing Phoenix-ECO cell lines. The SFG-m1928z-CD40L vector was constructed

by stepwise Gibson Assembly (New England BioLabs) using the cDNA of previously described anti-mouse CD19 scFv[49], Myc-tag sequence (EQKLISEEDL), murine CD28 transmembrane and an intracellular domain, murine CD3ζ intracellular domain without the stop codon, P2A self-cleaving peptide, and the murine CD40L protein.

**Mouse T cell isolation and retroviral transduction**. Mouse T cells were processed as described previously[6]. In brief, murine T cells were isolated from spleens of euthanized mice using the EasySep Mouse T cell Isolation Kit (StemCell). Cells were then expanded in RPMI-1640 supplemented with 10% heat-inactivated FBS, nonessential amino acids, 1 mM sodium pyruvate, 10 mM HEPES, 2 mM L-glutamine, 1% penicillin/streptomycin, 11 mM glucose, 2 μM 2-mercaptoethanol, 100 IU of recombinant human IL-2 (rhIL-2) (Prometheus Therapeutics & Diagnostics), and stimulated with anti-CD3/28 Dynabeads (Life Technologies) at a ratio of 1-to-2 (bead-to-cell). 24 and 48 h after bead stimulation, T cells were spinoculated on retronectin-coated plates with viral supernatant collected from Phoenix-ECO cells. After the second spinoculation, cells were rested for one day and then used in subsequent experiments.

**Adoptive transfer of CAR T cells**. For tumor studies, mice were inoculated i.v. with $1 \times 10^6$ firefly luciferase-expressing tumor cells on day 0. On day 6, bioluminescence imaging using the Xenogen IVIS Imaging System (Xenogen) with Living Image software (Xenogen) for the acquisition of imaging datasets was done to guarantee equal tumor burden of mice at time of treatment. Mice were then randomized into different treatment cohorts and on day 7, mice were treated with $1–3 \times 10^6$ CAR$^+$ T cells intravenously. Tumor burden over time was monitored by bioluminescent imaging and quantified over the whole animal body as photons/second/cm$^2$/steradian (p/s/cm$^2$/sr).

**Tumor challenges with CAR-antigen-negative tumor cells**. Long-term surviving BALB/c mice (50+ after initial tumor challenge with CD19$^+$ tumor cells) were inoculated i.v. with $1 \times 10^5$ A20.CD19-KO cells (BALB/c). Naive age-matched mice served as controls. Survival was monitored over time.

**Depletion of CD4$^+$ or CD8$^+$ cell populations**. To deplete CD4$^+$ T cells in tumor-bearing mice, 200 μg of anti-CD4 depletion antibody (GK1.5) or IgG control antibody (LTF-2) were injected i.p. 2x per week for 4 weeks starting one week prior to CAR T cell treatment. To deplete CD8$^+$ T cells in mice for re-challenge experiments with CD19-negative tumor cells, 200 μg of anti-CD8 depletion antibody (2.43) or IgG control antibody (LTF-2) were injected i.p. on days −3, 0, 7, 14, and 21 relative to tumor cell challenge. Depletion of CD4$^+$ and CD8$^+$ T cells was confirmed in the peripheral blood of treated mice by different antibody clones via flow cytometry (RM4-5 for CD4 and 53–6.7 for CD8 staining).

**Cell isolation for subsequent analyses**. Spleen and tumor tissue was processed as described previously[6]. Mice were euthanized via CO2 inhalation prior to organ removal. Harvested spleens were minced, filtered, washed in PBS, and red blood cells were lysed. Tumor tissue from the liver was mechanically disrupted, filtered, separated by Percoll density centrifugation, and red blood cells were lysed. Remaining cells were washed in PBS, counted, and used in subsequent analyses.

**Flow cytometry and FACS sorting**. Flow cytometric analyses were performed using a Beckman Coulter Gallios or a Thermo Fisher Attune NxT flow cytometer. Data were analyzed using FlowJo (Tree Star). DAPI (0.5 mg/ml, Sigma-Aldrich) or a LIVE/DEAD fixable violet dead cell stain kit (Thermo Fisher) were used to exclude dead cells in all experiments, and anti-CD16/CD32 antibody (93) was used to block non-specific binding of antibodies via Fc receptors. The following anti-mouse antibodies were used for flow cytometry: TruStain fcX (anti-mouse CD16/32) BioLegend Cat# 101319, RRID:AB_1574973, 5 μg/ml; anti-mouse CCR7 (clone 4B12) PE BioLegend, 120105, 2 μg/ml; anti-mouse CD3 (clone 17A2) BrilliantViolet510 BioLegend 100233, RRID:AB_2561387, 1 μg/ml; anti-mouse CD3ε (clone 145-2C11) PE-eFluor 610 eBioscience 61-0031, RRID:AB_2574251, 1 μg/ml; anti-mouse CD4 (GK1.5) AlexaFluor 700 eBioscience 56-0041, RRID:AB_493999, 0.1 μg/ml; anti-mouse CD8α (53-6.7) APC-eFluor 780 eBioscience 47-0081, RRID:AB_1272185, 0.1 μg/ml; anti-mouse/human CD11b (M1/70) AlexaFluor 700 eBioscience 56-0112, RRID:AB_657585), 0.1 μg/ml; anti-mouse CD11c (N418) APC-eFluor 780 eBioscience 47-0114, RRID:AB_1548663, 0.2 μg/ml; anti-mouse CD19 (eBio1D3) APC-eFluor 780 eBioscience 47-0193, RRID:AB_10853189, 0.1 μg/ml; anti-mouse CD19 (eBio1D3) PE eBioscience 12-0193, RRID:AB_657661, 0.1 μg/ml; anti-mouse CD19 (eBio1D3) PE-eFluor 610 eBioscience 61-0193, RRID: AB_2574536, 0.5 μg/ml; anti-mouse CD40 (1C10) PerCP-eFluor 710 eBioscience 46-0401, RRID:AB_2573677, 1 μg/ml; anti-mouse CD40L (MR1) PE eBioscience 12-1541, RRID:AB_465887, 0.2 μg/ml; anti-mouse CD45 (30-F11) BV605 BioLegend 103139, RRID:AB_2562341, 0.5 μg/ml; anti-mouse CD45 (30-F11) PE-Cy7 eBioscience 25-0451, RRID:AB_469625, 0.5 μg/ml; anti-mouse CD45.1 (A20) PE-eFluor610 eBioscience 61-0453, RRID:AB_2574560, 0.2 μg/ml; anti-mouse CD45.2 (104) PE-Cy7 eBioscience 25-0454, RRID:AB_2573350, 0.1 μg/ml; anti-mouse CD103 (2E7) BV711 BioLegend 121435, RRID:AB_2686970, 1 μg/ml; anti-mouse IFNγ (XMG1.2) PE-Cy7 eBioscience 25-7311, RRID:AB_1257211, 0.4 μg/ml; anti-

moue IRF8 (V3GYWCH) PerCP-eFluor710 eBioscience 46-9852, 1.6 μg/ml; anti-mouse Ki-67 (SolA15) PE-eFluor610 eBioscience 61-5698, 0.1 μg/ml; anti-mouse Ly-6G/Ly-6C (Gr-1) (RB6-8C5) PE-eFluor 610 eBioscience 61-5931, RRID: AB_2574639, 0.2 μg/ml; anti-mouse Ly-6G/Ly-6C (Gr-1) (RB6-8C5) PE-Cy7 eBioscience 25-5931, RRID:AB_469662, 0.2 μg/ml; anti-mouse MHC class II (MHC-II) I-A/I-E (M5/114.15.2) BV510 BioLegend 107635, RRID:AB_2561397, 0.2 μg/ml; anti-human Myc-tag (9B11) AlexaFluor 647 Cell Signaling 2233 S, RRID:AB_10693328, 1:500. Quantification of total cell numbers by flow cytometry was done using 123count eBeads Counting Beads (Thermo Fisher). For intracellular staining of IFNγ, a single cell suspension of tumor or spleen tissue was generated and cells were stimulated with 1x Cell Stimulation Cocktail (PMA, ionomycin, brefeldin A, and monensin) from Thermo Fisher for 5 h. Cells were then processed with the Cytofix/Cytoperm Plus kit (BD Biosciences) per the manufacturer's instructions. For intracellular IRF8 staining, the eBioscience Foxp3/Transcription Factor Staining Buffer Set (Thermo Fisher) was used according to the manufacturer's instructions. All antibodies were purchased from Biolegend, BD Biosciences, Cell Signaling, eBioscience, or Thermo Fisher. Sorting of splenocytes after tissue processing was done using a BD FACSAria under sterile conditions. Purity of cell populations was determined by reanalysis of an aliquot of sorted cell samples.

**Ex vivo DC culture assay**. Bone-marrow cultures were generated by seeding $1.2 \times 10^6$/ml CD45.1$^+$ BM cells into 24-well plates (500 μl/well) in complete RPMI media. Cells were supplemented with 100 ng/ml murine Flt3-Ligand (Peprotech) and 20 ng/ml murine GM-CSF (Peprotech). On day 2 of culture, $5–10 \times 10^3$ sorted CD45.2$^+$CD11b$^−$CD103$^−$ DN cDCs, CD11b$^−$CD103$^+$ cDC1, and CD11b$^+$CD103$^+$ cDC2 cells from CAR T cell-treated mice were separately added to the culture. After 3 days of co-culture, the percentage of DN cDCs, cDC1s, and cDC2s of all CD45.2$^+$ cells was assessed by flow cytometry.

**Quantification and statistical analysis**. All statistical analyses were performed using GraphPad Prism software (GraphPad). Data points represent biological replicates and are shown as the mean ± SEM or mean ± SD as indicated in the figure legends. Statistical significance was determined using an unpaired two-tailed Student's $t$-test, unless otherwise noted. The log-rank (Mantel-Cox) test was used to determine statistical significance for overall survival in mouse survival experiments. Significance was assumed with $*p < 0.05$; $**p < 0.01$; $***p < 0.001$; $****p < 0.0001$.

**Reporting summary**. Further information on research design is available in the Nature Research Reporting Summary linked to this article.

## Data availability
The authors declare that all data supporting the findings of this study are available within the paper and Supplementary Information. Source data are provided with this paper.

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

## Acknowledgements

We thank the Flow Cytometry, Molecular Cytology core for technical assistance; the Center for Experimental Therapeutics at Memorial Sloan Kettering Cancer Center (MSKCC) for innovations in structures, functions and targets of mAb-based drugs for cancer; Drs. A. Schietinger and M. Sadelain for excellent critical comments; A. Rookard (MSKCC CMG) for assistance in mouse breeding; and all members of the Brentjens laboratory for critical comments and discussion. We want to acknowledge the following funding sources: this work was supported by a National Cancer Institute fellowship 5F31CA213668-02 (N.F.K.), National Institutes of Health grants R01CA138738-05, PO1CA059350, PO1CA190174-01, and P50CA192937-03 (R.J.B.); The Annual Terry Fox Run for Cancer Research (New York, NY); Kate's Team; the Cabot Family Charitable Trust; the Leukemia and Lymphoma Society; the William Lawrence and Blanche Hughes Foundation; the Emerald Foundation (R.J.B.); and the institutional grant P30CACA008748 from the NIH.

## Author contributions

Conceptualization, N.F.K. and R.J.B.; Investigation, N.F.K., A.V.L., X.L., W.C., and A.F.D.; Writing – Original Draft, N.F.K.; Writing – Review & Editing, A.F.D. and R.J.B.; Supervision, R.J.B.; Funding Acquisition, N.F.K. and R.J.B.

## Competing interests

R.J.B. receives royalties and grant support from Juno Therapeutics, is a consultant for Juno Therapeutics/Celgene and Gracell Therapeutics, Inc., and has submitted a patent related to this work. The authors have no additional financial interests.
