## [Peer Review File · Nature Communications]

Reviewers' comments:

Reviewer #1, expert on CAR-T cells (Remarks to the Author):

Kuhn and colleagues report the results of a mouse study in which they have administered T-cells co-expressing a CD19 CAR and CD40L and show that ablation of the cDC1 subsets reduces the anti-tumor activity of adoptively transferred T-cells using the disseminated A20 lymphoma model. The authors also show the evidence of endogenous T-cell response directed at non-CD19 antigens which is boosted by the CD40L expression on CAR T-cells. Overall, the work is done at a high technical level but the following limitations severely limit interpretation of the results as well as interest to the general audience.

1. The report is an extension of the previously published work, in which the authors already described phenotypic changes in the DC subsets and stimulation of the endogenous CD8 T-cell response. For the full understanding/interpretation, readers have to read the prior report, and the current manuscript falls short as a standalone paper. The authors should include fundamental findings such as the enhanced anti-tumor activity, subset phenotype and the status of the endogenous T-cell response in the experiments shown in this report as separate panels. Otherwise the report appears orphaned and the significance unclear.

2. The authors conclude that the enhanced function of CD40L-expressing CAR T-cells is mediated in principle via the Batf3-dependent DC subset. However, the data does not fully support this conclusion as deletion of Batf3 (and the cDC1 subset) also severely impairs stimulation of endogenous CD8+ T-cell response (Fig. 3D, E) and erases any survival advantage (Fig. 2B) by the control CD19 CAR T-cells. This strongly indicates the cDC1 subset is critical for the anti-tumor function of CAR T-cells, regardless of the CD40L expression. In light of this, the main finding of this paper is that Batf3 expression in the host is critical for the priming/generation of endogenous CD8+ T-cell responses against immunogenic targets, likely by promoting the development of the cDC1 cell subset, and that this endogenous CD8+ T-cell response can be further boosted by CD40 stimulation. Unfortunately, this knowledge has been previously established in various studies of DC in mouse models, and therefore this report is confirmatory (now using a CAR T-cell model) rather than exploratory.

3. If we assume that the main contribution of Batf3 in this model is indeed the generation of cDC1, it is still unclear whether the other subset (cDC2) plays any role in stimulating the immune response. This is important because cDC2 is still the prevalent DC subset in both tumor and the spleen, as shown in Fig. 1. Would *Ltbr*^{-/-} mice show the same defect?

4. It is unclear how the CD19 CAR T-cells affect the cDC1/cDC2 balance. Do cDC2 convert to cDC1 in the presence of CAR T-cells? Do CD19 CAR T create an inflammatory environment that attracts cDC1 from LN or converts monocytes to cDC1? Do CD19 CAR T-cells stimulate proliferation of cDC1?

5. Lymphoma, especially when administered iv, does not usually establish solid masses with high stromal development and immunosuppressive environment as many bona fide solid tumors do. Therefore, it is unclear whether this mechanism will be observed in "real" solid tumor models.

Minor points:

1. Figure 2A should have statistical analysis in addition to the representative contour plots

2. In Fig 3, Student's t test cannot be used when several groups are compared. A one-way ANOVA should be used instead and corrected for multiple comparisons.

Reviewer #2, expert on tumor antigen presentation (Remarks to the Author):

The authors of the manuscript entitled "CD103+ Dendritic Cells and Endogenous CD8+ T Cells are Necessary for CD40 Ligand-Modified CAR T Cell Function" have shown previously that treatment with CD40L CAR T cells improves tumor control through direct CD40/CD40L mediated cytotoxicity and indirect induction of non-CAR T cell immunity that recognizes tumor cells. In the current study, the cell populations responsible for the induction of non-CAR CD8 T cell immunity is investigated in more detail and a role for Batf3-restricted DCs (cDC1) promoting this response is identified. This finding is not surprising given the extensive literature on the ability of cDC1 to cross-present antigens to the CD8 T cell compartment. The authors speculate the main CD8 T cell priming event occurs within the tumor tissue, but from the data presented there is no evidence to directly support this conclusion. While data presented indicates the endogenous tumor-specific T cell population generated may provide protective antitumor memory, alternative interpretations can be drawn from the limited experiments performed. Although I find the study interesting, there are major issues with the experimental design and interpretation of results versus the conclusions being drawn – many possible interpretations are present and should be carefully considered. The manuscript is preliminary, also very descriptive and does not provide any mechanistic insights.

1. Although the authors show increases in the proportion of cDC1 in the tumor following CD40L CAR T treatment, there is no evidence to suggest priming of the endogenous compartment is occurring within this site. Further experiments are warranted to draw this conclusion. Furthermore, why was analysis of the tumor-draining lymph node(s) excluded throughout the study? This would be the most logical site for CD8 T cell priming.
2. Statistical analysis in Figure 3E is not shown for the two treatment arms in Batf3^{-/-} mice. Elevated levels of tumor-specific CD8 T cells appear to be present in the CD40L CAR T group as compared to the control. How are these CD8 T cells being primed in the absence of cDC1?
3. Alternative explanations for the data presented in Figure 4 are possible. For example, protective immunity may be dependent on the presence of the CD40L CAR T cells during rechallenge – with the CAR T essential for promoting re-expansion post challenge and/or involved in direct CD40/CD40L killing. Experiments to exclude these possibilities should be performed: e.g. depletion of the CAR T cells by targeting the congenic marker and/or transfer of endogenous CD8 T cells to a new host prior to rechallenge.
4. . Are tumor-specific tissue-resident memory cells formed and if so, are they effectively depleted prior to rechallenge?
5. The data presented appears preliminary with some experiments having as few as 2 mice/group. Can reliable statistical analysis be performed on such a small sample size? Rechallenge experiments (Figure 4D) required 19 mice – why are only five mice shown in the initial treatment data? Have the investigators repeated any of these findings to demonstrate reproducibility?
6. How broad is the endogenous T cell response generated?
7. How are the transferred CD40L CAR T modulating the cDC1 compartment?
8. Why is the ratio of cDC1 to cDC2 different at differing sites – was this also observed in the tumor-

draining lymph nodes?

Reviewer #3, expert on CAR-T cells (Remarks to the Author):

In this manuscript Kuhn and colleagues describe a followup study from a recent manuscript (Kuhn et al, Cancer Cell, 2019) in which they demonstrate that coexpression of CD40-ligand on CAR T cells enhances their activity, decreases the needs for lymphodepletion, and increases the endogenous T cell response, allowing for elimination of antigen negative tumor cells. In the current manuscript, the authors use a BATF3- KO mouse model (which lacks type 1 conventional dendritic cells) in order to elucidate the mechanism of how overexpression of CD40L on syngeneic CAR T cells results in enhanced efficacy. While the KO model is interesting and does provide a small window of mechanistic insight to the previously reported finding of enhancement of the endogenous immune response, it does not explain all of the improved efficacy obtained by using the CD40L CAR T cells. Additionally, the previously published manuscript already showed that CD40L+ CAR T cells license dendritic cells (and that their increased activity was ablated in CD40 KO mice), thus it is not completely surprising that in this current manuscript that these CD40L+ CAR T cells prime CD8 cells. Overall, this paper is interesting but does not substantially add fundamentally new knowledge about the function or mechanism of CD40L+ CAR T cells.

Figure 1:

Figure B/C- This difference in DC recruitment by CAR T cells overexpressing CD40L vs those that do not to tumor vs peripheral lymphoid tissue was already previously shown in the last publication.

Figure D -What is new and nicely demonstrated here is that the makeup of the dendritic cell types is different for mice treated with CD40L+ CAR T cells with cDC1 being a larger proportion in the tumor and cDC2 being a larger proportion in the periphery. However, the authors do not dive into the larger questions of what this means. What is the role of cDC1 v cDC2 in the periphery? Do those matter or is this merely an observation?

Figure 2: This is a very small figure, can likely be combined with figure 1. The level of the effect here of BATF3-KO is small. The KM curves are somewhat similar (though there are a number of mice cured in WT and not BATF3-KO, those mice that do die of tumor do so at similar times). Additionally, the CD40L+ CAR T cells maintain greater activity compared to traditional CAR T cells even in BATF3-KO mice. Why is this? What is the mechanism other than dendritic cell priming of T cells? The authors go down the mechanism of the cDC1 priming, but this is only a small part of the mechanism of why CD40L+ CAR T cells are superior (and one that had been explored previously). Additionally, I wonder whether the BATF3-KO mice have cDC2? If so, are they still increased in the periphery when treated with CD40L+ CAR T cells?

Figure 3:

3A-B: Here, the authors first show that endogenous T cells obtained from mice treated with CD40L+ CAR T cells make more cytokine in response to PMA/Ionomycin stimulation than endogenous T cells obtained from mice treated with traditional CAR T cells in both WT and BATF3-KO mice. This would indicate that the CD40L+ CAR T cells somehow stimulate or prime the other T cells to be more effective-would be nice to look at this mechanism as it could account for most of the reason CD40L+ CAR T cells are superior to traditional CARs.

3C-E: They also used congenic markers to analyze the cytokine produced by endogenous T cells obtained from these mice in response to tumor. Here, they say that the endogenous cells in mice

treated with CD40L+ CARs are only superior to mice treated with regular CARs in those mice that are WT and not BATF3-KO. However, that does not appear to be supported by the data. In 3D, the ELISpots do appear to be more abundant in the bottom right than bottom left. Additionally, though it may not be statistically significant likely due to high variability, the numbers in 3E are clearly greater for CD40L+ CAR treated BATF3-KO mice than those treated with regular CAR T cells. Thus again, CD40L CAR T cells seem to have an effect on endogenous CAR T cells that is not dependent on cDC1, and this should be investigated more thoroughly. The more significant difference here seems to be that overall there is a decrease in cytokine produced by endogenous T cells from BATF3 KO mice. This may be due to cDC1 deletion, but BATF3-KO can have other effects on immune cells, so this should be confirmed after antibody depletion of cDC1.

Figure 4: I have no comments, this is well performed, but frankly not entirely surprising that CD8 depletion would prevent tumor rejection.

Reviewers' comments:

Reviewer #1, expert on CAR-T cells (Remarks to the Author):

Kuhn and colleagues report the results of a mouse study in which they have
administered T-cells co-expressing a CD19 CAR and CD40L and show that
ablation of the cDC1 subsets reduces the anti-tumor activity of adoptively
transferred T-cells using the disseminated A20 lymphoma model. The authors
also show the evidence of endogenous T-cell response directed at non-CD19
antigens which is boosted by the CD40L expression on CAR T-cells. Overall, the
work is done at a high technical level but the following limitations severely limit
interpretation of the results as well as interest to the general audience.

1. The report is an extension of the previously published work, in which the
authors already described phenotypic changes in the DC subsets and stimulation
of the endogenous CD8 T-cell response. For the full understanding/interpretation,
readers have to read the prior report, and the current manuscript falls short as a
standalone paper. The authors should include fundamental findings such as the
enhanced anti-tumor activity, subset phenotype and the status of the
endogenous T-cell response in the experiments shown in this report as separate
panels. Otherwise the report appears orphaned and the significance unclear.

We appreciate the comments of Reviewer #1 and agree by adding additional
data to the revised manuscript describing the immunophenotype of the cDC1 and
cDC2 subsets (Figure 1E and Supplementary Figure 2A-D) in spleen, primary
tumor tissue (=liver), and the tumor-draining lymph nodes (=coeliac & portal LNs;
(1)). We have taken the reviewer's advice and report the enhanced antitumor
activity of m1928z-CD40L CAR T cells in WT and *Batf3*^{-/-} mice as separate
panels (Fig. 2A and 2B). The decreased antitumor response of m1928z-CD40L
CAR T cells in cDC1-lacking *Batf3*^{-/-} mice is emphasized in a separate panel as
well (Fig. 2C). Also, as suggested, the status of both the endogenous and
adoptively transferred T cells is shown as separate panels in the revised
manuscript (Supplementary Figure 4).

2. The authors conclude that the enhanced function of CD40L-expressing CAR

T-cells is mediated in principle via the Batf3-dependent DC subset. However, the
data does not fully support this conclusion as deletion of Batf3 (and the cDC1
subset) also severely impairs stimulation of endogenous CD8+ T-cell response
(Fig. 3D, E) and erases any survival advantage (Fig. 2B) by the control CD19
CAR T-cells. This strongly indicates the cDC1 subset is critical for the anti-tumor
function of CAR T-cells, regardless of the CD40L expression. In light of this, the
main finding of this paper is that Batf3 expression in the host is critical for the
priming/generation of endogenous CD8+ T-cell responses against immunogenic
targets, likely by promoting the development of the cDC1 cell subset, and that
this endogenous CD8+ T-cell response can be further boosted by CD40
stimulation. Unfortunately, this knowledge has been previously established in
various studies of DC in mouse models, and therefore this report is confirmatory
(now using a CAR T-cell model) rather than exploratory.

We agree with the reviewer's comment that previous work has established the
importance of the cDC1 subset in various antitumor responses (2,3). Additionally,
previous reports have both shown the technical feasibility of therapeutically
enhancing the endogenous T cell antitumor response by pharmacologically
increasing the cDC1 numbers in the tumor tissue of preclinical mouse models
(4); as well as a positive correlation between immune checkpoint blockade
responses in cancer patients and cDC1 numbers in human tumor samples (5,6).
Thus, our report is focused on highlighting the feature of the armored CAR,
m1928z-CD40L, which combines the cytotoxic antitumor function of a CAR with
the ability of actively recruiting cDC1s to the tumor site in one treatment modality.
We would like to point out that the antitumor effect of control m1928z CAR T cells
is not affected by deletion of Batf3 (Fig. 2B).

Whereas previous reports from our lab have documented the improved
antitumor response and general stimulation of certain immune effectors, here, we
report the specific relevance of the cDC1-CD8 T cell axis in CD40L-armored
CAR-treated mice. Armored CAR T cells can optimize and have been
demonstrated to improve the antitumor response. Here we provide a mechanistic

insight as to how these CD40L-armed CAR T cells function. We have added
data and a complete figure (Figure 3) highlighting the effect of CD40L-armed
CAR T cells on the intratumoral conventional DC population: stimulation of
tumor-resident CD11b⁻ CD103⁻ double-negative (DN) cDCs to proliferate,
upregulate IRF8, and differentiate to cDC1s. Thus, we would like to emphasize
that this report goes beyond being just confirmatory and demonstrating how
CD40L-CAR T cells increase the intratumoral cDC1-to-cDC2 ratio.

3. If we assume that the main contribution of *Batf3* in this model is indeed the
generation of cDC1, it is still unclear whether the other subset (cDC2) plays any
role in stimulating the immune response. This is important because cDC2 is still
the prevalent DC subset in both tumor and the spleen, as shown in Fig. 1. Would
*Ltbr*^{-/-} mice show the same defect?

We acknowledge the reviewer's point that the importance of the cDC2 subset is
not directly assessed in our system. So far, evaluating the involvement of cDC2s
in the antitumor response has been challenging in the field. No equivalent
knockout mouse or other experimental tool currently exists that faithfully and
systemically only depletes the cDC2 population in mice. Whereas *Notch2*^{flox/flox}
*Itgax-cre* mice lack cDC2 cells in spleen and small intestine lamina propria, other
tissues are not depleted of cDC2s, and cDC1s also display a different
transcriptional profile when *Notch2* is knocked out in these mice (7,8). Similar
results were reported in mice lacking the transcription factor *Irf4*, which is
necessary for proper cDC2 development. Genetic ablation of *Irf4* in mice
generally decreased cDC2 numbers and impaired their function to migrate to
lymph nodes, but did not completely ablate them systemically (9,10). Besides
*Batf3*^{-/-} mice, which specifically affect the development of one immune cell
subtype (=cDC1; (3)), other knockout mice are warranted to assess the
involvement of other DC subtypes in antitumor responses.

We appreciate the reviewer's suggestion of using *Ltbr*^{-/-} mice. Mice lacking the
lymphotoxin beta receptor have a defective secondary lymphoid compartment,

do not develop lymph nodes, have disorganized splenic B cell follicles, and
defective DC homeostasis (11,12). Thus, one would not be able to attribute any
potential antitumor defect in *Ltbr*^{-/-} mice to a specific DC subtype. We have added
a paragraph regarding cDC2 depletion in the results section.

4. It is unclear how the CD19 CAR T-cells affect the cDC1/cDC2 balance. Do
cDC2 convert to cDC1 in the presence of CAR T-cells? Do CD19 CAR T create
an inflammatory environment that attracts cDC1 from LN or converts monocytes
to cDC1? Do CD19 CAR T-cells stimulate proliferation of cDC1?

We would like to thank the reviewer for this comment and investigated the
impact of CAR T cell treatment on the cDC1/cDC2 balance. This additional
analysis was added to the revised manuscript as a separate figure (Figure 3).

This question prompted us to assess how the different CAR T cell treatments
affect the cDC subpopulations. Common dendritic cell precursors (CDPs) in the
bone marrow differentiate to recently identified “pre-cDC1s” and “pre-cDC2s”
(13,14). Schlitzer et al. could show that isolated pre-cDC1s and pre-cDC2s from
the bone marrow specifically differentiated to mature cDC1s and cDC2s,
respectively, in the periphery after pre-cDC transfer into a naïve host (13). This
inspired us to analyze any potential changes in CDP, pre-cDC1, and pre-cDC2
populations in the bone marrow (=site of DC-poiesis) of CAR T cell treated mice,
which would potentially explain the changes we see in the periphery. However,
both m1928z and m1928z-CD40L CAR T cell-treated mice had unchanged CDP
and pre-cDC populations in the bone marrow (data not shown). A recently
published report using adoptively transferred T cells expressing Flt3L showed
that numbers of bone marrow-resident pre-cDCs can be therapeutically
increased, resulting in higher numbers of CD103⁺ DCs in the tumor (15).

In contrast, our findings suggested that any changes in cDC1/cDC2 ratios we
see in the periphery, are stimulated independently of pre-cDC development in the
bone marrow. Focusing on the peripheral, differentiated cDC populations, we
next assessed the expression of the IRF8 transcription factor in tumor-derived

cDC populations. In the periphery, IRF8 controls survival and function of
terminally differentiated cDC1s (16,17). Furthermore, increased IRF8 expression
in CD11b⁻ CD103⁻ double-negative (DN) cells was shown to promote their
differentiation into mature CD103⁺ cDC1s (4). Thus, we hypothesized that
CD40L-CAR T cell treatment skews the cDC1/cDC2 ratio towards the cDC1
populations by stimulating the DN cells to expand, upregulate IRF8, and
differentiate into cDC1s. We specifically noticed upregulation of IRF8 (readout of
DN-to-cDC1 differentiation) and Ki-67 (readout for proliferation) in DN cells
treated with CD40L-CAR T cell-treated mice (Fig 3B). The increased expression
of Ki-67 in the tumor-derived DN cells also correlated with the observed increase
of the DN population in the tumor of CD40L-CAR T cell-treated mice (Fig 1F),
indicating that DN cells receive a proliferative signal upon CD40L-CAR T cell
treatment. Intriguingly, splenic DN cells and DN cells from the tdLNs did not
upregulate Ki-67 or IRF8 (Figures 3C and 3D), implying a tumor-specific effect.

Next, we wanted to assess if IRF8 upregulation in the DN population leads to
DN-to-cDC1 differentiation. To address this, DN cells were isolated by FACS
from tumors of m1928z and m1928z-CD40L CAR T cell-treated mice and
cultured ex vivo for 3 days to assess their potential to differentiate to cDC1s
without any further stimuli. Both, DN cells from m1928z and m1928z-CD40L CAR
T cell-treated mice differentiated into cDC1s ex vivo, albeit DN cells from
m1928z-CD40L CAR T cell-treated mice differentiated 2x more efficiently
compared to m1928z CAR T cell-treated mice (Figure 3E). Together, this
suggests that CD40L-CAR T cells affect the intratumoral cDC1/cDC2 ratio by
stimulating CD11b⁻ CD103⁻ DN cell proliferation, upregulation of the cDC1-
skewing IRF8 transcription factor, and, consequently, differentiation of DN cDCs
to cDC1s in the tumor tissue.

Additionally, to address the reviewer's question about cDC1-to-cDC2 trans-
differentiation, we did not detect any IRF8 upregulation in cDC2s (data not
shown). However, in the ex vivo culture system, a small percentage of cDC2s
(~1/6th) did give rise to cDC1s (Supplementary Figure 3B), suggesting that this

trans-differentiation is possible. This was observed in DN populations of both
m1928z and m1928z-CD40L CAR T cell-treated mice, indicating that this effect is
not specific to either CAR T cell treatment cohort.

Also, regarding the question of proliferation of cDC populations after CAR
treatment, Ki-67 staining showed that, both, cDC1s and cDC2s proliferated more
after CD40L-CAR T cell treatment (Figures 3F and 3G). Thus, CD40L-CAR T
cells do not specifically stimulate the cDC1 population, but instead affect the
intratumoral DN progenitors. Why this increased proliferation of cDC subsets in
the tumor does not translate to an increase in overall numbers (Figure 1C), is
addressed in the discussion section of the revised manuscript.

5. Lymphoma, especially when administered iv, does not usually establish solid
masses with high stromal development and immunosuppressive environment as
many bona fide solid tumors do. Therefore, it is unclear whether this mechanism
will be observed in "real" solid tumor models.

We agree with the reviewer's comment that our lymphoma model does not
recapitulate a bona fide immunosuppressive TME. Pairing the CD40L platform
with a CAR targeting a solid tumor in a syngeneic mouse model is warranted to
address this question but beyond the scope of this manuscript. The presented
data is still relevant to current CAR T cell trials, as clinical data using non-
armored anti-CD19 CAR T cells in B cell malignancies requires further
improvement.

Minor points:

1. Figure 2A should have statistical analysis in addition to the representative
contour plots

We have added the statistical analysis matching the representative contour
plots.

2. In Fig 3, Student's t test cannot be used when several groups are compared. A
one-way ANOVA should be used instead and corrected for multiple comparisons.

We thank the reviewer for pointing out the correct statistical analysis. We have
revised the figure accordingly. The cytokine stimulation data can now be found
under Supplementary Figure 4, whereas the ELIspot data was removed.

Reviewer #2, expert on tumor antigen presentation (Remarks to the Author):

The authors of the manuscript entitled “CD103+ Dendritic Cells and
Endogenous CD8+ T Cells are Necessary for CD40 Ligand-Modified CAR T Cell
Function” have shown previously that treatment with CD40L CAR T cells
improves tumor control through direct CD40/CD40L mediated cytotoxicity and
indirect induction of non-CAR T cell immunity that recognizes tumor cells. In the
current study, the cell populations responsible for the induction of non-CAR CD8
T cell immunity is investigated in more detail and a role for Batf3-restricted DCs
(cDC1) promoting this response is identified. This finding is not surprising given
the extensive literature on the ability of cDC1 to cross-present antigens to the
CD8 T cell compartment. The authors speculate the main CD8 T cell priming
event occurs within the tumor tissue, but from the data presented there is no
evidence to directly support this conclusion. While data presented indicates the
endogenous tumor-specific T cell population generated may
provide protective antitumor memory, alternative interpretations can be drawn
from the limited experiments performed. Although I find the study interesting,
there are major issues with the experimental design and interpretation of results
versus the conclusions being drawn – many possible interpretations are present
and should be carefully considered. The manuscript is preliminary, also very
descriptive and does not provide any mechanistic insights.

1. Although the authors show increases in the proportion of cDC1 in the tumor
following CD40L CAR T treatment, there is no evidence to suggest priming of the
endogenous compartment is occurring within this site. Further experiments are
warranted to draw this conclusion. Furthermore, why was analysis of the tumor-
draining lymph node(s) excluded throughout the study? This would be the most
logical site for CD8 T cell priming.

We appreciate the reviewer’s suggestion to analyze the tumor-draining lymph
nodes (tdLNs) and have included data reporting differences in m1928z and
m1928z-CD40L CAR T cell-treated mice in the revised manuscript.

As primary tumor growth occurs in the liver after i.v. injection of A20 lymphoma
cells, we focused our analysis on the coeliac and portal LNs, which drain the liver
tissue in mice (1). The changes in cDC subtype ratio in the tdLN mirrored the
results seen in the spleen of m1928z-CD40L CAR T cell-treated mice (spleen:
Fig 1G; tdLN: Supplementary Fig 1F), suggesting that the effect of the CD40L-
armored CAR is consistent across secondary lymphoid tissues. Similar to spleen,
cDC1s in both the migratory (CD11b- CD103+) and resident (CD11b- CD8a+)
DC compartment were not the dominant population when mice received m1928z-
CD40L CAR T cells (Supplementary Figure 1E and 1F). Furthermore, migDN
DCs in the tumor-draining LN of CD40L-CAR T cell treated mice were not
stimulated to proliferate (measured by Ki-67 staining) or expressed higher levels
of IRF8 (an indicator for DN-to-cDC1 differentiation; Fig. 3D). These results were
consistent with the spleen (Fig 3C; see also Reviewer #1, Comment & Response
#4), whereas DN DCs in the tumor expressed higher levels of the proliferation
marker Ki-67 and IRF8 in m1928z-CD40L CAR T cell treated mice (Fig 3B).

The tdLN and spleen also shared additional similarities: m1928z-CD40L CAR T
cell treatment increased recruitment of DCs to both anatomical sites (Fig 1C and
Fig 1D). In the tdLN, the migDC population (identified by MHC-II^{hi} CD11c^{mid}
expression) outnumbered the resDC population (MHC-II^{low} CD11c^{hi}) when mice
received m1928z-CD40L CAR T cells (Fig 1D). The increased recruitment of
migDC into the tdLN of these mice was supported by the higher CCR7
expression on tumor-resident DCs (Fig 1E), a chemokine receptor binding
CCL19 and CCL21, which are predominantly produced in LNs and mediate
homing of lymphoid and myeloid cells to the LN (18). Thus, m1928z-CD40L CAR
T cells lead to increased recruitment of DCs into secondary lymphoid organs
(tdLN & spleen) and their activation (Figures 1C, 1D, and Supplementary Figure
2A; (19)), but this is not a systemic effect, as the liver (as the primary tumor site)
is not infiltrated by more DCs upon treatment.

2. Statistical analysis in Figure 3E is not shown for the two treatment arms in

Batf3^{-/-} mice. Elevated levels of tumor-specific CD8 T cells appear to be present
in the CD40L CAR T group as compared to the control. How are these CD8 T
cells being primed in the absence of cDC1?

We thank the reviewer for pointing out the missing statistical analysis. For
clarity, we have elected to remove Figures 3C to 3E from the original manuscript.
We have repeated the ELIspot experiments originally described in Figures 3C to
3E and observed the same trends as seen in the original manuscript, however,
not to a statistically significant degree. We now consider our initial experimental
ELIspot protocol not sufficient to show endogenous CD8 T cell priming in our
system. Due to the recent COVID-19-related lab shutdown, we are currently not
able to explore alternative experiments.

Cross-priming of CD8 T cells independently of Batf3-expressing cDC1s has
been described. CD169⁺ macrophages (20,21) have been identified as possible
antigen crosspresenters for CD8 T cell stimulation in LNs. In our system, we
have previously reported the activation of both macrophages and DCs (19),
warranting further work to establish a potential stimulatory role of macrophages
in CAR T cell-treated mice. The lack of Batf3-expressing cDC1s impairs the
m1928z-CD40L CAR T cell antitumor response (Fig 2D). Identification and
depletion of other cross-presenting cells could possibly completely ablate the
antitumor response and provide evidence that other non-cDC1s are involved as
well. This comment was added to the discussion section.

3. Alternative explanations for the data presented in Figure 4 are possible. For
example, protective immunity may be dependent on the presence of the CD40L
CAR T cells during rechallenge – with the CAR T essential for promoting re-
expansion post challenge and/or involved in direct CD40/CD40L killing.
Experiments to exclude these possibilities should be performed: e.g. depletion of
the CAR T cells by targeting the congenic marker and/or transfer of endogenous
CD8 T cells to a new host prior to rechallenge.

We agree with the reviewer's concern regarding an alternative explanation to
the findings in Figure 4 and attempted to address the potential CD40/CD40L
killing of residual CD40L-CAR T cells with the following experiment:

To exclude the possibility that any residual CD40L-CAR T cells in long-term
surviving mice target A20.CD19-KO cells via CD40/CD40L-directed cytotoxicity,
we collected long-term surviving mice that had normal levels of peripheral B cells
(see Figure A, below). This indicated that these mice had no circulating functional
anti-CD19 CAR T cells anymore, because B cell aplasia in humans and mice is a
readout for the presence of functional anti-CD19 CAR T cells (22,23).
Additionally, in this second re-challenge experiment, we used A20 CD40 and
CD19 double-knock out cells (A20.CD40-CD19.DKO), further excluding the
possibility that if there potentially are circulating non-functional m1928z-CD40L
CAR T cells, the tumor cells would not be susceptible to CD40/CD40L-mediated
cytotoxicity. One out of the 5 anti-CD8 depleted mice did survive and had no
tumor growth (20% survival), whereas 2 out of 5 of the IgG control mice did
succumb to tumor re-challenge (60% survival). Thus, a statistical significance
between the two groups is not reached (see Figure B, below).

Due to the long nature of this experiment (50+ days for generating long-term
surviving mice, plus 50+ days for the re-challenge and CD8-depletion part), in
combination with the recent COVID-19-related lab shutdown, we were not able to
repeat this experiment and have not included this data set in the manuscript.
Whereas this preliminary result is promising in suggesting that CD40/CD40L-
mediated cytotoxicity is not protective in long-term surviving mice, we
acknowledge increased sample numbers are necessary to draw a conclusion.

**Figure A.** Relative counts of CD19+ B cells in the peripheral blood of age-
 matched and long-term surviving mice. Long-term survivors do not present with B
 cell aplasia, a biomarker for anti-CD19 CAR T cell persistence. p-value was
 determined by unpaired Student's t-test. ns, non-significant.

**Figure B.** Survival of mice treated with CD8 T cell-depleting antibody (clone
 2.43) or non-depleting IgG control antibody). Naïve age-matched BALB/c mice
 were used as controls. All p-values are were determined by a log-rank (Mantel
 Cox) test.

4. Are tumor-specific tissue-resident memory cells formed and if so, are they
effectively depleted prior to rechallenge?

We acknowledge that we do not know if tissue-resident memory cells are
formed. If there are T_{RM} CD8 T cells present and they are not depleted by anti-
CD8a antibody treatment, these T_{RM} CD8 T cells are not sufficient to protect mice
from tumor re-challenge (Fig. 4).

5. The data presented appears preliminary with some experiments having as
few as 2 mice/group. Can reliable statistical analysis be performed on such a
small sample size? Rechallenge experiments (Figure 4D) required 19 mice – why
are only five mice shown in the initial treatment data? Have the investigators
repeated any of these findings to demonstrate reproducibility?

We thank the reviewer for pointing out the limited number of sample size in
certain experiments and have updated the revised manuscript to reflect more
reliable statistical analysis. The interpretation of the data in question remains
unchanged.

Also, we have included additional survival graphs with long-term surviving
m1928z-CD40L CAR T cell-treated mice in the Supplemental Figure 5. These
long-term surviving mice were used in subsequent re-challenge experiments and
were collected from independently performed experiments to demonstrate
experimental reproducibility.

6. How broad is the endogenous T cell response generated?

We acknowledge that we have not quantified the degree of endogenous T cell
clones responding to the tumor challenge. However, we would like to emphasize
that Figure 4 demonstrates overall depletion of CD8 T cells prevents protection
from tumor re-challenge. TCR sequencing and/or flow cytometry-based TCR V β
analysis of the T cell repertoire upon re-challenge with antigen-negative tumor
cells, as done in that experiment, could provide evidence for the clonality of the
protective T cell response.

7. How are the transferred CD40L CAR T modulating the cDC1 compartment?

As Reviewer #1 has asked a similar question, we have copied our response
here and hope that it satisfies this critique:

We appreciate the reviewer's suggestion to analyze the tumor-draining lymph
nodes (tdLNs) and have included data reporting differences in m1928z and
m1928z-CD40L CAR T cell-treated mice in the revised manuscript.

As primary tumor growth occurs in the liver after i.v. injection of A20 lymphoma
cells, we focused our analysis on the coeliac and portal LNs, which drain the liver
tissue in mice (1). The changes in cDC subtype ratio in the tdLN mirrored the
results seen in the spleen of m1928z-CD40L CAR T cell-treated mice (spleen:
Fig 1G; tdLN: Supplementary Fig 1F), suggesting that the effect of the CD40L-
armored CAR is consistent across secondary lymphoid tissues. Similar to spleen,
cDC1s in both the migratory (CD11b⁻ CD103⁺) and resident (CD11b⁻ CD8a⁺)
DC compartment were not the dominant population when mice received m1928z-
CD40L CAR T cells (Supplementary Figure 1E and 1F). Furthermore, migDN
DCs in the tumor-draining LN of CD40L-CAR T cell treated mice were not
stimulated to proliferate (measured by Ki-67 staining) or expressed higher levels
of IRF8 (an indicator for DN-to-cDC1 differentiation; Fig. 3D). These results were
consistent with the spleen (Fig 3C; see also Reviewer #1, Comment & Response
#4), whereas DN DCs in the tumor expressed higher levels of the proliferation
marker Ki-67 and IRF8 in m1928z-CD40L CAR T cell treated mice (Fig. 3B).

The tdLN and spleen also shared additional similarities: m1928z-CD40L CAR T
cell treatment increased recruitment of DCs to both anatomical sites (Fig 1C and
Fig 1D). In the tdLN, the migDC population (identified by MHC-II^{hi} CD11c^{mid}
expression) outnumbered the resDC population (MHC-II^{low} CD11c^{hi}) when mice
received m1928z-CD40L CAR T cells (Fig 1D). The increased recruitment of
migDC into the tdLN of these mice was supported by the higher CCR7
expression on tumor-resident DCs (Fig 1E), a chemokine receptor binding
CCL19 and CCL21, which are predominantly produced in LNs and mediate

homing of lymphoid and myeloid cells to the LN (18). Thus, m1928z-CD40L CAR
T cells lead to increased recruitment of DCs into secondary lymphoid organs
(tdLN & spleen) and their activation (Figures 1C, 1D, and Supplementary Figure
2A; (19)), but this is not a systemic effect, as the liver (as the primary tumor site)
is not infiltrated by more DCs upon treatment.

8. Why is the ratio of cDC1 to cDC2 different at differing sites – was this also
observed in the tumor-draining lymph nodes?

As outlined in Response #1, we have now included data of the tdLN in the
revised manuscript.

A discussion of different cDC1-to-cDC2 ratios in different tissues was added to
the Discussion section:

“Why the cDC1-to-cDC2 ratio increases in tumors of m1928z-CD40L CAR T
cell treated mice is unclear and warrants further investigation. The accumulation
of cDC1s in the tumor tissue has been attributed to several NK cell-derived
cytokines such as CCL5, FLT3L, and XCL1 (5,6). Conventional DCs in peripheral
tissue have a half-life of about 3 to 6 days and are maintained by tissue-resident
pre-cDCs that originate in and exit from the bone marrow (24,25). This process
can be observed in a mouse model of influenza infection, when pre-cDCs traffic
to the infected lung tissue and locally increase the cDC numbers (26). We see
increased proliferation of cDCs after CD40L-CAR T cell treatment only in the
tumor and not in lymphoid tissue. More importantly, CD40L-CAR T cell treatment
skews the cDC1-to-cDC2 ratio in favor of the cDC1s by promoting differentiation
of progenitor IRF8⁺ DN progenitor cells to cDC1s. This is similar to published
results, where homeostasis and generation of cDCs in peripheral tissue is
maintained by mobilization of progenitor cDCs from the bone marrow (4,26). Both
endogenous and exogenously applied FLT3L are instructive in mediating this
effect (4,24,25), suggesting a pathway that potentially is activated upon CD40L-
CAR T cell treatment. It is unclear if pre-cDCs found in different tissues respond
to differentiation signals differently, warranting further analysis of progenitor DCs

residing in different tissues. “

Reviewer #3, expert on CAR-T cells (Remarks to the Author):

In this manuscript Kuhn and colleagues describe a followup study from a recent
manuscript (Kuhn et al, Cancer Cell, 2019) in which they demonstrate that
coexpression of CD40-ligand on CAR T cells enhances their activity, decreases
the needs for lymphodepletion, and increases the endogenous T cell response,
allowing for elimination of antigen negative tumor cells. In the current manuscript,
the authors use a BATF3- KO mouse model (which lacks type 1 conventional
dendritic cells) in order to elucidate the mechanism of how overexpression of
CD40L on syngeneic CAR T cells results in enhanced efficacy. While the KO
model is interesting and does provide a small window of mechanistic insight to
the previously reported finding of enhancement of the endogenous immune
response, it does not explain all of the improved efficacy obtained by using the
CD40L CAR T cells. Additionally, the previously published manuscript already
showed that CD40L+ CAR T cells license dendritic cells (and that

their increased activity was ablated in CD40 KO mice), thus it is not completely
surprising that in this current manuscript that these CD40L+ CAR T cells prime
CD8 cells. Overall, this paper is interesting but does not substantially add
fundamentally new knowledge about the function or mechanism of CD40L+ CAR
T cells.

Figure 1:

Figure B/C- This difference in DC recruitment by CAR T cells overexpressing
CD40L vs those that do not to tumor vs peripheral lymphoid tissue was already
previously shown in the last publication.

Figure D -What is new and nicely demonstrated here is that the makeup of the
dendritic cell types is different for mice treated with CD40L+ CAR T cells with
cDC1 being a larger proportion in the tumor and cDC2 being a larger proportion

in the periphery. However, the authors do not dive into the larger questions of
what this means. What is the role of cDC1 v cDC2 in the periphery? Do those
matter or is this merely an observation?

The role of cDC1 versus cDC2 in the periphery is currently of great scientific
and translational interest, as both cell populations interact with CD8 and CD4 T
cells, respectively, to instruct immune responses against pathogens, as well as
malignant cell growth (27). In the periphery, cDC1s sample tumor material,
upregulate CCR7 to migrate to draining LNs, where they are the most potent
CD8 T cell stimulators, compared to other DC subtypes (4,28). Direct priming of
CD8 T cells by cDC1s independently of LN migration has also been described
(29), indicating that cDC1s in the periphery are essential to initiate an antitumor
CD8 T cell response.

As also mentioned above in response to Reviewer #1 Comment #3, the role of
peripheral cDC2s in the antitumor response is less explored. This can be
attributed to the lack of faithful cDC2-depletion methods, both genetic and
pharmacologic methods can only partially deplete cDC2s or inhibit their migratory
potential (27). Thus, we are limited in assessing the relevance of cDC2s in our
system. This concern was added to the results section of the revised manuscript.

Focusing on the role of peripheral cDC1s in our model, we can show that their
relative accumulation compared to cDC2s is specifically induced by the CD40L-
armored CAR T cells (Fig 1H and 1I). Additionally, their presence is necessary
for the CD40L-armored CAR T cells to exert their full antitumor effect (Fig 2A and
2D). Furthermore, in *Cd40*^{-/-} mice, which do not benefit from CD40L-armored
CAR treatment (19), the changes in peripheral DC subtypes is not observed,
indicating a connection between improved antitumor response, CD40-CD40L
host interactions, and peripheral cDC1 accumulation (Figure 1I and
Supplementary Figure 2E). This new data was added to the revised manuscript.

Figure 2: This is a very small figure, can likely be combined with figure 1. The
level of the effect here of BATF3-KO is small. The KM curves are somewhat

similar (though there are a number of mice cured in WT and not BATF3-KO,
those mice that do die of tumor do so at similar times). Additionally, the CD40L+
CAR T cells maintain greater activity compared to traditional CAR T cells even in
BATF3-KO mice. Why is this? What is the mechanism other than dendritic cell
priming of T cells? The authors go down the mechanism of the cDC1 priming, but
this is only a small part of the mechanism of why CD40L+ CAR T cells are
superior (and one that had been explored previously).

Additionally, I wonder whether the BATF3-KO mice have cDC2? If so, are they
still increased in the periphery when treated with CD40L+ CAR T cells?

We thank the reviewer for pointing out cDC1-independent antitumor
mechanisms that are induced by CD40L-armed CAR T cells. A similar point of
discussion was raised by Reviewer #2 and we discuss cross-presentation of
antigens to CD8 T cells independently of cDC1s as another mechanism for T cell
priming under “Reviewer #2 Comment #2”. Additionally, we want to emphasize
that we do not propose that the improved antitumor effect of CD40L-armed
CAR T cells is solely dependent on cDC1-CD8 T cell priming. As presented in
Figure 4, both endogenous T cells and, more importantly, CAR T cells in
m1928z-CD40L-treated mice produce more IFN γ effector cytokine, even when
cDC1s are absent in *Batf3*^{-/-} mice. Other *Cd40* expressing cells, such as
macrophages and non-cDC1s, can be licensed by CD40L-armed CAR T cells,
provide a pro-inflammatory environment, and thereby aid the CD40L+ CAR T
cells in an improved antitumor response (19). This was added to the discussion
section.

*Batf3*^{-/-} mice are selectively depleted of cDC1s (3) (Fig 2A). They still have
cDC2s. The absolute number of peripheral cDC2s in both mice is unchanged
after m1928z-CD40L CAR T cell treatment:

Figure 3:

3A-B: Here, the authors first show that endogenous T cells obtained from mice
 treated with CD40L+ CAR T cells make more cytokine in response to
 PMA/Ionomycin stimulation than endogenous T cells obtained from mice treated
 with traditional CAR T cells in both WT and BATF3-KO mice. This would indicate
 that the CD40L+ CAR T cells somehow stimulate or prime the other T cells to be
 more effective-would be nice to look at this mechanism as it could account for
 most of the reason CD40L+ CAR T cells are superior to traditional CARs.

For clarity, this data is now found as Supplementary Figure 4.

As pointed out in the previous response to Figure 2, m1928z-CD40L CAR T
 cells induce licensing of both splenic cDC1 (Supplementary Figure 2A) and non-
 cDC1 myeloid cell populations (19). Subsequently, we demonstrated that host
 *Cd40* expression is necessary for CD40-CD40L crosstalk between host myeloid
 cells and CD40L+ CAR T cells, as this effect of myeloid cell licensing is lost in
 *Cd40*^{-/-} mice. Concurrently, endogenous T cells are also not primed in *Cd40*^{-/-}
 mice when treated with CD40L+ CAR T cells and these mice are not able to
 mount an effective antitumor immune response (19). Thus, we attribute the
 production of IFN γ and TNF α of endogenous T cells after non-specific
 PMA/Ionomycin stimulation in the context of CD40L+ CAR T cell treatment to
 host *Cd40* expression, and not to the presence of cDC1s.

3C-E: They also used congenic markers to analyze the cytokine produced by
 endogenous T cells obtained from these mice in response to tumor. Here, they

say that the endogenous cells in mice treated with CD40L+ CARs are only
superior to mice treated with regular CARs in those mice that are WT and not
BATF3-KO. However, that does not appear to be supported by the data. In 3D,
the ELISpots do appear to be more abundant in the bottom right than bottom left.
Additionally, though it may not be statistically significant likely due to high
variability, the numbers in 3E are clearly greater for CD40L+ CAR treated
BATF3-KO mice than those treated with regular CAR T cells. Thus again, CD40L
CAR T cells seem to have an effect on endogenous CAR T cells that is not
dependent on cDC1, and this should be investigated more thoroughly. The more
significant difference here seems to be that overall there is a decrease in
cytokine produced by endogenous T cells from BATF3 KO mice. This may be
due to cDC1 deletion, but BATF3-KO can have other effects on immune cells, so
this should be confirmed after antibody depletion of cDC1.

We thank the reviewer for pointing out the difference in IFN γ cytokine
production in *Batf3*^{-/-} mice treated with CAR or CD40L+ CAR T cells. For clarity,
we have elected to remove Figures 3C to 3E from the original manuscript. We
have repeated the ELIspot experiments originally described in Figures 3C to 3E
and observed the same trends as seen in the original manuscript, however, not
to a statistically significant degree. We now consider our initial experimental
ELIspot protocol not sufficient to show endogenous CD8 T cell priming in our
system. Due to the recent COVID-19-related lab shutdown, we are currently not
able to explore alternative experiments.

cDC1-independent CD8 T cell antigen-crosspriming has been described in the
context of antitumor responses. CD169+ macrophages have been identified as
possible crosspresenters for CD8 T cell stimulation (21). Further work is
warranted to identify this cell type in our system and a possible significance in
cDC1-independent cross-presentation. However, any changes in T cell priming
seen in *Batf3*^{-/-} mice when compared to wild-type mice can be attributed to the
absence of cDC1s. Hildner et al. demonstrated that knocking out the transcription
factor *Batf3* specifically depleted the cDC1 population, whereas other immune

cell populations (B cells, CD4 T cells, CD8 T cells, NK cells, cDC2s,
plasmacytoid DCs) were not affected (3). We are not aware of antibody-mediated
cDC1 depletion, as cDC1s do not express a surface marker that is exclusively
expressed by cDC1s (for example, integrin alpha E (= CD103) is also expressed
by tissue-resident memory T_{RM} cells (30)). We have added a paragraph
addressing non-cDC1-mediated CD8 T cell crosspriming to the discussion
section of the revised manuscript.

Figure 4: I have no comments, this is well performed, but frankly not entirely
surprising that CD8 depletion would prevent tumor rejection.

References:

- 1. Barbier L, Tay SS, McGuffog C, Triccas JA, McCaughan GW, Bowen DG,
et al. Two lymph nodes draining the mouse liver are the preferential site of
DC migration and T cell activation. *J Hepatol.* 2012;
- 2. Broz ML, Binnewies M, Boldajipour B, Nelson AE, Pollack JL, Erle DJ, et
al. Dissecting the Tumor Myeloid Compartment Reveals Rare Activating
Antigen-Presenting Cells Critical for T Cell Immunity. *Cancer Cell.* 2014;
- 3. Hildner K, Edelson BT, Purtha WE, Diamond M, Matsushita H, Kohyama
618 M, et al. Batf3 deficiency reveals a critical role for CD8 α + dendritic cells in
cytotoxic T cell immunity. *Science* (80-). 2008;
- 4. Salmon H, Idoyaga J, Rahman A, Leboeuf M, Remark R, Jordan S, et al.
Expansion and Activation of CD103+ Dendritic Cell Progenitors at the
Tumor Site Enhances Tumor Responses to Therapeutic PD-L1 and BRAF
Inhibition. *Immunity.* 2016;44(4):924–38.
- 5. Barry KC, Hsu J, Broz ML, Cueto FJ, Binnewies M, Combes AJ, et al. A
natural killer–dendritic cell axis defines checkpoint therapy–responsive
tumor microenvironments. *Nat Med.* 2018;
- 6. Böttcher JP, Bonavita E, Chakravarty P, Blees H, Cabeza-Cabrerizo M,
Sammicheli S, et al. NK Cells Stimulate Recruitment of cDC1 into the
Tumor Microenvironment Promoting Cancer Immune Control. *Cell.* 2018;

- 7. Lewis KL, Caton ML, Bogunovic M, Greter M, Grajkowska LT, Ng D, et al.
Notch2 receptor signaling controls functional differentiation of dendritic
cells in the spleen and intestine. *Immunity*. 2011;
- 8. Satpathy AT, Briseño CG, Lee JS, Ng D, Manieri NA, Kc W, et al. Notch2-
dependent classical dendritic cells orchestrate intestinal immunity to
attaching-and-effacing bacterial pathogens. *Nat Immunol*. 2013;
- 9. Schlitzer A, McGovern N, Teo P, Zelante T, Atarashi K, Low D, et al. IRF4
Transcription Factor-Dependent CD11b+ Dendritic Cells in Human and
Mouse Control Mucosal IL-17 Cytokine Responses. *Immunity*. 2013;
- 10. Bajaña S, Turner S, Paul J, Ainsua-Enrich E, Kovats S. IRF4 and IRF8 Act
in CD11c + Cells To Regulate Terminal Differentiation of Lung Tissue
Dendritic Cells . *J Immunol*. 2016;
- 11. Fütterer A, Mink K, Luz A, Kosco-Vilbois MH, Pfeffer K. The lymphotoxin β
receptor controls organogenesis and affinity maturation in peripheral
lymphoid tissues. *Immunity*. 1998;
- 12. Kabashima K, Banks TA, Ansel KM, Lu TT, Ware CF, Cyster JG. Intrinsic
lymphotoxin- β receptor requirement for homeostasis of lymphoid tissue
dendritic cells. *Immunity*. 2005;
- 13. Schlitzer A, Sivakamasundari V, Chen J, Sumatoh HR Bin, Schreuder J,
Lum J, et al. Identification of cDC1- and cDC2-committed DC progenitors
reveals early lineage priming at the common DC progenitor stage in the
bone marrow. *Nat Immunol*. 2015;
- 14. Grajales-Reyes GE, Iwata A, Albring J, Wu X, Tussiwand R, Kc W, et al.
Batf3 maintains autoactivation of Irf8 for commitment of a CD8 α +
conventional DC clonogenic progenitor. *Nat Immunol*. 2015;
- 15. Lai J, Mardiana S, House IG, Sek K, Henderson MA, Giuffrida L, et al.
Adoptive cellular therapy with T cells expressing the dendritic cell growth
factor Flt3L drives epitope spreading and antitumor immunity. *Nat*
*Immunol*. 2020;
- 16. Sichien D, Scott CL, Martens L, Vanderkerken M, Van Gassen S, Plantinga
660 M, et al. IRF8 Transcription Factor Controls Survival and Function of

- Terminally Differentiated Conventional and Plasmacytoid Dendritic Cells,
Respectively. *Immunity*. 2016;
- 17. Ginhoux F, Liu K, Helft J, Bogunovic M, Greter M, Hashimoto D, et al. The
origin and development of nonlymphoid tissue CD103⁺ DCs. *J Exp Med*.
2009;
- 18. Förster R, Davalos-Miszlitz AC, Rot A. CCR7 and its ligands: Balancing
immunity and tolerance. *Nature Reviews Immunology*. 2008.
- 19. Kuhn NF, Purdon TJ, van Leeuwen DG, Lopez A V., Curran KJ, Daniyan
AF, et al. CD40 Ligand-Modified Chimeric Antigen Receptor T Cells
Enhance Antitumor Function by Eliciting an Endogenous Antitumor
Response. *Cancer Cell*. 2019;
- 20. Bernhard CA, Ried C, Kochanek S, Brocker T. CD169⁺ macrophages are
sufficient for priming of CTLs with specificities left out by cross-priming
dendritic cells. *Proc Natl Acad Sci U S A*. 2015;
- 21. Asano K, Nabeyama A, Miyake Y, Qiu CH, Kurita A, Tomura M, et al.
CD169-Positive Macrophages Dominate Antitumor Immunity by
Crosspresenting Dead Cell-Associated Antigens. *Immunity*. 2011;
- 22. Brentjens RJ, Rivière I, Park JH, Davila ML, Wang X, Stefanski J, et al.
Safety and persistence of adoptively transferred autologous CD19-targeted
T cells in patients with relapsed or chemotherapy refractory B-cell
leukemias. *Blood*. 2011;
- 23. Davila ML, Kloss CC, Gunset G, Sadelain M. CD19 CAR-Targeted T Cells
Induce Long-Term Remission and B Cell Aplasia in an Immunocompetent
Mouse Model of B Cell Acute Lymphoblastic Leukemia. *PLoS One*.
2013;8(4).
- 24. Liu K, Victora GD, Schwickert TA, Guermonprez P, Meredith MM, Yao K,
et al. In Vivo Analysis of Dendritic Cell Development and Homeostasis.
*Science* (80-). 2009;
- 25. Hochweller K, Miloud T, Striegler J, Naik S, Hämmerling GJ, Garbi N.
Homeostasis of dendritic cells in lymphoid organs is controlled by
regulation of their precursors via a feedback loop. *Blood*. 2009;

- 26. Cabeza-Cabrerizo M, van Blijswijk J, Wienert S, Heim D, Jenkins RP,
Chakravarty P, et al. Tissue clonality of dendritic cell subsets and
emergency DCpoiesis revealed by multicolor fate mapping of DC
progenitors. *Sci Immunol*. 2019;
- 27. Durai V, Murphy KM. Functions of Murine Dendritic Cells. *Immunity*. 2016.
- 28. Roberts EW, Broz ML, Binnewies M, Headley MB, Nelson AE, Wolf DM, et
al. Critical Role for CD103+/CD141+ Dendritic Cells Bearing CCR7 for
Tumor Antigen Trafficking and Priming of T Cell Immunity in Melanoma.
*Cancer Cell*. 2016;
- 29. Thompson ED, Enriquez HL, Fu Y-X, Engelhard VH. Tumor masses
support naive T cell infiltration, activation, and differentiation into effectors.
*J Exp Med*. 2010;
- 30. Masopust D, Soerens AG. Tissue-Resident T Cells and Other Resident
Leukocytes. *Annu Rev Immunol*. 2019;

REVIEWER COMMENTS

Reviewer #1 (Remarks to the Author):

While the first version of the manuscript was somewhat underwhelming, the authors made an impressive effort to address key concerns, which made the paper much stronger.

My only remaining request is that the authors acknowledge in the Discussion section the limitations of using a disseminated lymphoma model in lieu of a bona fide solid tumor (Query #5).

MM

Reviewer #2 (Remarks to the Author):

The authors have provided additional information within the revised manuscript to clarify several points raised previously.

1. There still remains no clear evidence on the location of T cell priming in the data presented. The authors include additional data suggesting migratory DCs on D7 are preferentially recruited to the draining nodes following treatment with CD40L CAR T cells (Fig1D). These migratory DCs are identified by increased expression of MHC II. An alternative interpretation is that intact CD40L signalling activates LN resident DCs and this could be why the proportion of MHC II^{high} DCs are present. Sole reliance on MHC II upregulation is not a reliable marker in these circumstances for identifying migratory DCs. In addition, were DCs enumerated in these experiments, consistent with the data presented in Fig 1 B-C? 2. The authors state treatment with CD40L CAR T skew the tumor resident cDC1 to cDC2 in favor of cDC1 – however, this is not observed in lymphoid compartments, where TDLN have decrease in CD103 migratory DC (Supp Fig 1E) and splenic cDC1 (Supp Fig 1D D7). These observations between tissues are not highlighted in the results, as well as the author's interpretations of this presented data. Is there any evidence that the increased recruitment of DCs across anatomical sites results in improved priming of T cells?

2. As this study focusses on the mechanisms underpinning CD40L-CAR T treatment, the role of non-cDC1 should be explored in light of the findings in Batf3 KO mice. Are these non-cDC1 cells playing a major role in the therapeutic efficacy observed following transfer of CD40L CAR T in wildtype mice.

3. An increase in sample size is necessary to draw an appropriate conclusion with the new data presented in the rebuttal.

6. Experiments showing increased breadth of endogenous T cells following CD40L CAR T treatment as suggested by the authors would strengthen the manuscript.

Reviewer #3 (Remarks to the Author):

The authors have address most of my concerns through additional experiments or in their discussion. They have done a nice job of discussing that non cDC1 cross-priming may be responsible for some of the enhanced anti-tumor activity in their model (and thus cDC1 are not solely responsible for the

activity).

I have one important concern:

Figure 2D appears to be repeated data from Figures 2B and 2C which is non-standard. The authors do not indicate how many times the experiments were performed (in this figure or others). This needs to be clarified as it appears that figure 2 is only from one experiment.

Reviewer #1 (Remarks to the Author):

While the first version of the manuscript was somewhat underwhelming, the authors made an impressive effort to address key concerns, which made the paper much stronger.

My only remaining request is that the authors acknowledge in the Discussion section the limitations of using a disseminated lymphoma model in lieu of a bona fide solid tumor (Query #5).

MM

Response:

We have edited the Discussion to acknowledge the limitations of our disseminated lymphoma model in comparison to solid tumor models – especially their differences in stromal involvement and immunosuppressive tumor microenvironments.

Reviewer #2 (Remarks to the Author):

The authors have provided additional information within the revised manuscript to clarify several points raised previously.

1. There still remains no clear evidence on the location of T cell priming in the data presented. The authors include additional data suggesting migratory DCs on D7 are preferentially recruited to the draining nodes following treatment with CD40L CAR T cells (Fig1D). These migratory DCs are identified by increased expression of MHC II. An alternative interpretation is that intact CD40L signalling activates LN resident DCs and this could be why the proportion of MHC II^{high} DCs are present. Sole reliance on MHC II upregulation is not a reliable marker in these circumstances for identifying migratory DCs. In addition, were DCs enumerated in these experiments, consistent with the data presented in Fig 1 B-C?

The lack of CD86 upregulation on the migratory DCs after m1928z-CD40L treatment suggests that LN-resident DCs do not receive an activation signal that explains the increase of the MHCII^{hi} CD11c^{int} cell fraction. Whereas CD86 was upregulated in splenic cDCs upon m1928z-CD40L CAR T cell treatment (Supplementary Figure 2A), the migDC subsets and the migDC as a whole did not change CD86 expression (Supplementary Figure 2D and E). We also did not detect any proliferation marker Ki67⁺ LN-resident cDCs (Supplementary Figure 2G), further suggesting that the increase in LN-resident migDC fraction in m1928z-CD40L CAR T cell-treated mice is not due to local proliferation.

The CD86 data on the whole splenic cDC population and the whole tdLN cDC population, as well as the Ki-67⁺ staining of tdLN cDCs was added to Supplementary Figure 2 and is referenced in the manuscript.

The DCs in tdLNs were not enumerated. As we have previously published, intravenous injection of the A20 lymphoma cell line leads to tumor nodule growth in the liver. The coeliac and portal lymph nodes have been identified as the liver-draining lymph nodes in the mouse (Barbier et al. 2012). We noticed that Balb/c mice did not universally present with both lymph nodes. This observation was not dependent on CAR T cell treatment (m1928z vs. m1928z-CD40L) and, therefore, we did not select to enumerate total cell numbers in the tdLNs.

2. The authors state treatment with CD40L CAR T skew the tumor resident cDC1 to cDC2 in favor of cDC1 – however, this is not observed in lymphoid compartments, where TDLN have decrease in CD103 migratory DC (Supp Fig 1E) and splenic cDC1 (Supp Fig 1D D7). These observations between tissues are not highlighted in the results, as well as the author's interpretations of this presented data. Is there any evidence that the increased recruitment of DCs across anatomical sites results in improved priming of T cells?

We acknowledge that we do not have an explanation for the observed discrepancy of cDC1-to-cDC2 ratios between tissue sites. m1928z-CD40L CAR T cell treatment did induce upregulation of the proliferative marker Ki-67 and the cDC1-differentiation marker IRF8 in CD11b⁻ CD103⁻ DN cDCs specifically in the tumor (Figure 3). This suggests that m1928z-CD40L CAR T cells skew the cDC1-to-cDC2 ratio by stimulating DN cDCs to predominantly differentiate to cDC1s. Why this is not observed in tdLNs or in the spleen, remains to be explored. Whereas the differentiation axis of progenitor DCs coming from the bone marrow is appreciated (Schlitzer et al. 2015), how they seed the different peripheral tissues and if they then respond to different stimulus cues is still subject of intense research (Cabeza-Cabrero et al. 2019). These discrepancies, along with the potential effect on T cell priming,

are mentioned in paragraph #3 in the Results section and discussed in paragraphs 5 and 6 in the Discussion section.

2. As this study focusses on the mechanisms underpinning CD40L-CAR T treatment, the role of non-cDC1 should be explored in light of the findings in Batf3 KO mice. Are these non-cDC1 cells playing a major role in the therapeutic efficacy observed following transfer of CD40L CAR T in wildtype mice.

We agree that CD40L-CAR T cell treatment perhaps licenses other non-cDC1 myeloid cells that directly contribute to the increased antitumor efficacy. We do not know if these non-cDC1 cells play a major role in the therapeutic efficacy of CD40L-armed CAR T cells in wildtype mice. In the current manuscript, we are focusing on the involvement of Batf3-expressing cDC1s and their necessity for observing the full potential of the CD40L-armed CAR T cell antitumor effect. We agree that the presence and contribution of other non-cDC1 cells is likely and needs to be explored. We have toned down our statement in the abstract from claiming that CD40L-armed CAR T cells lose their antitumor function, to stating that CD40L-armed CAR T cells “elicit an impaired antitumor response”. We felt that this better reflects the point raised by the reviewer and the results shown in Figure 2.

Candidates for further evaluation are other cross-presenting cells, such as CD169+ macrophages in the peripheral tissue and lymph nodes, blood-circulating monocytes, or cDC2s. *Irf4*^{-/-} mice provide non-complete depletion of cDC2s with impaired function to migrate to lymph nodes (Bajaña et al. 2016; Schlitzer et al. 2013). Adapting our current model from Balb/c to C57BL/6 mice will enable us to utilize such genetic mouse models on the C57BL/6 background. This is planned in future studies.

The possible involvement of other non-cDC1 myeloid cells in CD40L-CAR T cell treatment is addressed in paragraph 4 of the Discussion section.

3. An increase in sample size is necessary to draw an appropriate conclusion with the new data presented in the rebuttal.

We agree with the reviewer that an increase in sample size is necessary in order to draw an appropriate and satisfactory conclusion, regarding a potential protective effect of residual m1928z-CD40L CAR T cells in this re-challenge experiment.

We would like to point out that we have so far never been able to experimentally demonstrate a protective CD40/CD40L-mediated killing effect of CD40L-armed CAR T cells *in vivo*. In our previous published work, the CD40/CD40L-directed killing was only observed *in vitro* when CD40L-armed CAR T cells were co-cultured at a 1-to-1 ratio with tumor cells (Supplementary Figure S1C&D in (Kuhn et al. 2019)). In the same study, non-tumor-recognizing CD40L-armed CAR T cells were not able to improve survival of mice (4h11m28mz-CD40L CAR T cells in Figure 2D (Kuhn et al. 2019)). Additionally, as shown in this study in Supplementary Figure S5A, tumor-recognizing non-cytotoxic CD4⁺ m1928z-CD40L-armed CAR T cells are also not capable of delaying tumor outgrowth in our tumor model. Thus, we would argue that the sole presence of CD40L-armed CAR T cells is not sufficient to induce protective CD40/CD40L-killing *in vivo*.

6. Experiments showing increased breadth of endogenous T cells following CD40L CAR T treatment as suggested by the authors would strengthen the manuscript.

We agree with the reviewer that our analysis does not encompass the elucidation of the breadth of the endogenous T cell response. Whereas antibody-mediated systemic depletion of CD8⁺ T cells abrogates the protective effect of CD40L-armed CAR T cells in re-challenge experiments, we have not identified the clonality of the putative endogenous CD8⁺ T cell response. This caveat is acknowledged in paragraph 2 of the Discussion section.

Reviewer #3 (Remarks to the Author):

The authors have address most of my concerns through additional experiments or in their discussion. They have done a nice job of discussing that non cDC1 cross-priming may be responsible for some of the enhanced anti-tumor activity in their model (and thus cDC1 are not solely responsible for the activity).

I have one important concern:

Figure 2D appears to be repeated data from Figures 2B and 2C which is non-standard. The authors do not indicate how many times the experiments were performed (in this figure or others). This needs to be clarified as it appears that figure 2 is only from one experiment.

Response:

We agree with the Reviewer's concern about Figure 2 and confirm that Figure 2D is data repeated from Figures 2B and 2C, highlighting the difference in m1928z-CD40L CAR T cell treated mice in the different mouse strains (WT vs Batf3^{-/-}). This was initially done upon request by Reviewer #1 (Query #1).

We have now changed it back to the initial graph (first submission) and the conventional way of plotting the data as one summary graph. The data is the summary of two independent experiments (see Source Data). Figure legends indicate how many times the experiments were performed. We thank the reviewer for pointing this out and have made the necessary changes in the revised manuscript.

References:

- Bajaña, Sandra et al. 2016. "IRF4 and IRF8 Act in CD11c + Cells To Regulate Terminal Differentiation of Lung Tissue Dendritic Cells ." *The Journal of Immunology*.
- Barbier, Louise et al. 2012. "Two Lymph Nodes Draining the Mouse Liver Are the Preferential Site of DC Migration and T Cell Activation." *Journal of Hepatology*.
- Cabeza-Cabrerizo, Mar et al. 2019. "Tissue Clonality of Dendritic Cell Subsets and Emergency DCpoiesis Revealed by Multicolor Fate Mapping of DC Progenitors." *Science Immunology*.
- Kuhn, Nicholas F. et al. 2019. "CD40 Ligand-Modified Chimeric Antigen Receptor T Cells Enhance Antitumor Function by Eliciting an Endogenous Antitumor Response." *Cancer Cell*.
- Schlitzer, Andreas et al. 2013. "IRF4 Transcription Factor-Dependent CD11b+ Dendritic Cells in Human and Mouse Control Mucosal IL-17 Cytokine Responses." *Immunity*.
- Schlitzer, Andreas et al. 2015. "Identification of CDC1- and CDC2-Committed DC Progenitors Reveals Early Lineage Priming at the Common DC Progenitor Stage in the Bone Marrow." *Nature Immunology*.